# Linear Time Algorithms for $k$-means with Multi-Swap Local Search

**Junyu Huang**[1,2]**, Qilong Feng**[1,2,*]**, Ziyun Huang**[3]**, Jinhui Xu**[4]**, Jianxin Wang**[1,2,5,*]

[1]School of Computer Science and Engineering, Central South University,
Changsha 410083, China
[2]Xiangjiang Laboratory, Changsha 410205, China
[3]Department of Computer Science and Software Engineering, Penn State Erie,
The Behrend College
[4]Department of Computer Science and Engineering, State University of New York at Buffalo,
NY, USA
[5]The Hunan Provincial Key Lab of Bioinformatics, Central South University,
Changsha 410083, China
`junyuhuangcsu@foxmail.com, csufeng@mail.csu.edu.cn,`
`zxh201@psu.edu, jinhui@buffalo.edu, jxwang@mail.csu.edu.cn`

## Abstract

The local search methods have been widely used to solve the clustering problems. In practice, local search algorithms for clustering problems mainly adapt the single-swap strategy, which enables them to handle large-scale datasets and achieve linear running time in the data size. However, compared with multi-swap local search algorithms, there is a considerable gap on the approximation ratios of the single-swap local search algorithms. Although the current multi-swap local search algorithms provide small constant approximation, the proposed algorithms tend to have large polynomial running time, which cannot be used to handle large-scale datasets. In this paper, we propose a multi-swap local search algorithm for the $k$-means problem with linear running time in the data size. Given a swap size $t$, our proposed algorithm can achieve a $(50(1 + \frac{1}{t}) + \epsilon)$-approximation, which improves the current best result 509 (ICML 2019) with linear running time in the data size. Our proposed method, compared with previous multi-swap local search algorithms, is the first one to achieve linear running time in the data size. To obtain a more practical algorithm for the problem with better clustering quality and running time, we propose a sampling-based method which accelerates the process of clustering cost update during swaps. Besides, a recombination mechanism is proposed to find potentially better solutions. Empirical experiments show that our proposed algorithms achieve better performances compared with branch and bound solver (NeurIPS 2022) and other existing state-of-the-art local search algorithms on both small and large datasets.

## 1   Introduction

Clustering is a fundamental problem in the field of machine learning with many real-world applications. The goal of clustering is to partition a given set of data points into different clusters according to their similarity such that data points within the same cluster share high similarity as much as possible. Among different objective functions, the $k$-means clustering aims to minimize the sum of the squared distances between data points to their closest centers. More formally, given a set $P \subset \mathbb{R}^d$

---

*Corresponding Authors

37th Conference on Neural Information Processing Systems (NeurIPS 2023).

of data points in a $d$-dimensional Euclidean space, the goal of the $k$-means clustering is to find a set $C \subset \mathbb{R}^d$ of size at most $k$ with the following objective: $\min_{C \subset \mathbb{R}^d} \sum_{p \in P} \min_{c \in C} \|p - c\|^2$.

For the $k$-means problem, Lloyd's algorithm [12] is one of the most widely used heuristic in practice. However, there is no theoretical guarantee for Lloyd-type method unless certain data distribution assumptions are introduced. It is known that there are several constant approximation schemes based on primal-dual and randomized rounding techniques [2, 5, 8]. The current best approximation ratio in polynomial time is 5.912 [5], which is based on primal-dual method and nested quasi-independent set. For fixed dimension $d$ or the number of clusters $k$, several $(1 + \epsilon)$-approximation algorithms were proposed [6, 7].

The $k$-means++ algorithm proposed by Arthur and Vassilvitskii [3] is a good seeding method that runs in linear time in the data size with $O(\log k)$-approximation. It is also known that the $k$-means++ algorithm gives a constant approximation by opening $O(k)$ centers [1, 13, 16]. Lattanzi and Sohler [11] showed a combination of $k$-means++ seeding and the local search algorithm (named as LS++ algorithm), which yields a constant approximation with running time $O(ndk^2 \log \log k)$. In each round, LS++ algorithm samples a data point using $k$-means++ seeding and enumerates possible swap pairs to make improvements on clustering cost. They proved that after $O(k \log \log k)$ rounds of sampling and swaps, one can obtain a 509-approximate solution in expectation. Choo et al. [4] proved that one can achieve an $O(1/\epsilon^3)$-approximation using reduced $O(\epsilon k)$ rounds of LS++ algorithm. Under the assumption that each optimal cluster has size $\Omega(n/k)$, Huang et al. [9] gave an improved approximation algorithm with ratio $(100 + \epsilon)$ by random sampling methods.

However, there are still several issues for local search methods. Although current local search methods with multi-swap strategy can achieve good theoretical guarantee, the running time of them have polynomial dependence on the data size, which are hard to be used to handle large-scale datasets. Compared with the $(9 + \epsilon)$-approximation multi-swap local search algorithm given in [10], the approximation ratio 509 of LS++ is a large constant since it can only apply the single-swap strategy. Numerical experiments [9] showed that LS++ could easily fall into a poor local optimum when handling real-world datasets. An immediate idea is to apply the multi-swap strategy to LS++ algorithm for improvements. However, the swapping process of LS++ relies heavily on the one-to-one matched swap pairs defined in [11]. Thus, it is challenging to apply the multi-swap strategy to solve the $k$-means problem while maintaining a linear dependence of data size on the running time.

Secondly, in the process of local search swaps, the time and space complexities for clustering cost update during swaps have great impact on the efficiency of the algorithms. A direct way in [11] is to maintain the nearest and the second nearest centers for each data point such that picking the best swap pair and updating the clustering cost can be implemented in time $O(nd)$ and $O(ndk)$, respectively. Maintaining the distances from data points to their centers requires an extra space complexity of $O(nd)$. To obtain faster implementation, as pointed out in [4], one can use binary search trees to store the distances from each data point to each of the clustering centers. By using this data structure, each local search step can be implemented in time $O(nd \log k)$. However, the space complexity becomes $O(ndk)$. To get much practical algorithms for the $k$-means problem, it is necessary to further improve the time and space complexities for updating the clustering cost during swaps.

## 1.1 Our Contributions

In order to further narrow the gap between theory and practice, in this paper, we propose the first multi-swap local search algorithm for the $k$-means problem with linear running time in the data size. A common feature for the existing multi-swap local search methods is that $\Theta((nk)^t)$ candidate swaps should be enumerated for finding clustering cost improvements in a single local search step, which leads to at least quadratic running time. To overcome this challenge, our idea is to use a sampling-based strategy to construct a candidate set of centers that are close to the optimal clustering centers in linear time, which serves as the set of potentially good centers for swapping in. Based on the candidate set of centers constructed, enumerations on the current set of centers opened suffice to determine a good swap. Hence, the number of possible swaps can be reduced from $\Theta((nk)^t)$ to $\Theta(k^t)$, where each local search step can be conducted in linear-time in the data size.

Sampling-based strategy has been used in [11] for designing single-swap local search algorithms. However, the theoretical analysis relies heavily on the one-to-one matched swap pairs. Thus, the approximation ratio is a large constant since there may exist some optimal clusters that cannot be well

approximated by performing matched swaps. In this paper, to obtain better approximation guarantee, we extend the notions of swap pairs and propose a new consecutive sampling method to construct candidate centers for swap such that data points close to a subset of optimal clustering centers can be swapped in simultaneously. A key challenge here is that there may exist some optimal clusters whose clustering costs only take a tiny fraction of the total clustering cost such that sampling methods may fail. To overcome this challenge, we propose new structures that divide optimal clusters into different groups for establishing a lower bound for the success probability of sampling.

By using the proposed multi-swap local search method (denoted as MLS algorithm for short), given a swap size $t$, an improved $(50(1 + \frac{1}{t}) + \epsilon)$-approximation can be obtained in time $O(ndk^{2t+1}\log(\epsilon^{-1}\log k))$. To benefit more from the proposed multi-swap local search method when handling large-scale datasets, we propose a sampling-based method for accelerating the MLS algorithm. The proposed algorithm (denoted as MLSP algorithm) accelerates the updating of clustering cost during the swaps such that each local search iteration in MLS algorithm can be implemented in time $O(ndk + poly(k)d)$ with extra space complexity of $\tilde{O}(kd)$. In order to obtain better clustering quality, we develop a recombination mechanism in MLSP which combines sampling and scoring methods to help the local search algorithm find better solutions when the search falls into a poor local optimum. By picking the top-$k$ data points with the highest scores as new initialization, the chance to get out of the local optimal solutions becomes large. Numerical experiments show that our proposed method achieves better performances compared with branch and bound solver and other local search algorithms. The main contributions of this paper are as follows.

- We propose the first multi-swap local search algorithm (MLS algorithm) with running time linearly dependent on the data size. Given a swap size $t$ with $t \geq 2$, our MLS algorithm achieves a $(50(1 + \frac{1}{t}) + \epsilon)$-approximation in time $O(ndk^{2t+1}\log(\epsilon^{-1}\log k))$, which improves the current best approximation ratio 509 with linear running time in the data size.

- We give a practical heuristic algorithm (MLSP algorithm) for better implementation of the proposed MLS algorithm, which accelerates the process of clustering cost update during swaps and provides better scalability to large-scale datasets. Besides, a recombination mechanism is proposed to prevent the local search algorithm falling into a poor local optimum too early.

Table 1 shows a detailed comparison of our results with the state-of-the-art ones.

| Result | Approximation Guarantee | Method | Assumption | Running Time |
|---|---|---|---|---|
| [3] | $O(\log k)$ | $k$-means++ | - | $O(ndk)$ |
| [10] | $(3 + \frac{2}{t})^2$ | Multi-Swap Local Search | - | $O(n^{t+1}k^t d \log \Delta)$ |
| [11] | 509 | Sampling + Single-Swap Local Search | - | $O(ndk^2 loglogk)$ |
| [4] | $O(1)$ | Sampling + Single-Swap Local Search | - | $O(ndk \log k)$ |
| [9] | $100 + \epsilon$ | Sampling + Single-Swap Local Search | $|P_h^*| \geq \frac{n\epsilon}{k}$ | $O(ndk^2 \log \epsilon^{-1})$ |
| This Paper | $50(1 + \frac{1}{t}) + \epsilon$ | Sampling + Multi-Swap Local Search | - | $O(ndk^{2t+1}\log(\epsilon^{-1}\log k))$ |

Table 1: Comparison with related results on $k$-means clustering, where $n$ is the size of the given dataset, $d$ is the dimension, $k$ is the number of clusters opened, $t$ is the parameter representing the swap size of local search methods, and $\Delta$ is the aspect ratio (aspect ratio is defined as the maximum pairwise distance of the given instance divided by the minimum pairwise distance).

## 2  Preliminaries

We use $P \subset \mathbb{R}^d$ and $k$ to denote the given dataset and the number of clusters, respectively. For any two points $p, q \in P$, we use $d(p, q) = \|p - q\|^2$ to denote the squared distance between them. Given two sets $A, B \subseteq P$, let $\Delta(A, B) = \sum_{p \in A} \min_{q \in B} d(p, q)$ denote the sum of the squared distances from data points in $A$ to their closest points in $B$. Let $C^* = \{c_1^*, c_2^*, ..., c_k^*\}$ be an optimal solution. We use $\mathbb{P}(C^*) = \{P_1^*, P_2^*, ..., P_k^*\}$ to denote the corresponding optimal clusters by assigning data points in $P$ to their closest centers in $C^*$. Let $Opt$ be the cost of the optimal solution. For a subset $Q \subseteq \mathbb{P}(C^*)$ of optimal clusters, we use $Z(Q) = \cup_{P_h^* \in Q} P_h^*$ to denote the set of data points in clusters of $Q$. Denote $Z'(Q) = \{c_h^* : P_h^* \in Q\}$ as the set of optimal centers of clusters in $Q$. Given a subset $S \subseteq C^*$, let $J(S) = \{P_h^* : c_h^* \in S\}$ be the collection of optimal clusters whose clustering centers are in $G$. For an integer $t$, let $[t] = [1, 2, ..., t]$. The following lemma is a folklore for the $k$-means problem.

**Lemma 1** [3] Let $P \subseteq \mathbb{R}^d$ be a set of data points, and $\mu(P) = \frac{1}{|P|}\sum_{p \in P} p$ denote the center of gravity. For any data point $c \in \mathbb{R}^d$, we can get $\Delta(P, \{c\}) = |P|d(\mu(P), c) + \Delta(P, \{\mu(P)\})$.

**Theorem 1** [3] Algorithm 1 returns an $O(\log k)$-approximate solution in time $O(ndk)$.

---

**Algorithm 1** $k$-means++

---

**Input**: An instance $(P, k)$ of the $k$-means problem.
**Output**: A set $C \subseteq \mathbb{R}^d$ of centers with size at most $k$.
  1: Randomly sample a point $p \in P$ and set $C = \{p\}$.
  2: **for** $i = 1$ to $k - 1$ **do**
  3:     Pick a point $p \in P$ with probability $\Delta(\{p\}, C)/\Delta(P, C)$, and add $p$ to $C$.
  4: **return** $C$.

---

## 3 Linear Time Local Search Algorithm with Multi-Swap Strategy

The general idea solving the $k$-means problem with multi-swap local search is that we propose a new consecutive sampling method to construct candidate centers for swap such that data points close to a subset of optimal clustering centers can be swapped in simultaneously. The multi-swap local search algorithm (denoted as MLS) is given in Algorithm 2. There are mainly two stages in each round of the MLS algorithm. Given a swap size $t$, in the first stage (steps 3 to 5), a candidate set of centers with size $t$ will be sampled using the $k$-means++ method. This avoids enumerating all the data points for constructing the candidate sets for swapping in. In the second stage (steps 6 to 7), the algorithm enumerates all subsets (with size at most $t$) of the candidate set of centers opened for swapping out. By extending the notions of swap pairs to swap set and carefully analyzing the structures of local optimal solutions, we prove that the clustering cost can be reduced significantly with certain probability in each iteration of the MLS Algorithm. The following is the main result of this paper.

**Theorem 2** *In the $i$-th iteration of Algorithm 2, let $C'$ be the set of centers obtained in step 7. If the current clustering cost $\Delta(P, C)$ is larger than $50(1 + \frac{1}{t})Opt$, then with probability at least $\Omega(k^{-t})$, we have $\Delta(P, C') \leq (1 - \frac{1}{100k})\Delta(P, C)$. After $O(k^{O(t)} \log(\epsilon^{-1} \log k))$ iterations, we get an approximate solution with ratio $(50(1 + \frac{1}{t}) + \epsilon)$ in expectation[2].*

### 3.1 Analysis

In this subsection, we analyze our proposed Algorithm 2, where a candidate set of centers for swap is constructed by $t$ independent sampling steps in each iteration. Our objective is to show that the clustering cost can be reduced with certain probability in each iteration. Due to space limit, all the detailed proofs are given in Appendix A.

In the following, we consider a single iteration of the proposed MLS algorithm. Assume $\Delta(P, C) \geq 50(1 + \frac{1}{t})Opt$ holds within a single iteration in Algorithm 2. Otherwise, $C$ is already a $50(1 + \frac{1}{t})$-approximate solution for $P$. Let $C = \{c_1, c_2, ..., c_k\}$ denote the set of centers before the swap (steps 6-7) of Algorithm 2. Let $\mathbb{P}(C) = \{P_1, P_2, ..., P_k\}$ be the corresponding partition of clusters induced by $C$. For an optimal cluster $P_h^*$, let $c_h^*$ be its clustering center. For each cluster $P_h \in \mathbb{P}(C)$, let $c_h \in C$ denote its clustering center. Following the ones in [11], we extend the definition of good clusters with respect to $C^*$ as follows.

**Definition 1** *Good single cluster.* *A cluster $P_h^* \in \mathbb{P}(C^*)$ is called good with a pair of points $(c_h^*, c_j)$ such that $c_j \in C$ and $\Delta(P_h^*, C) - \zeta(P, C, c_h^*, c_j) - 9\Delta(P_h^*, \{c_h^*\}) > \frac{1}{100k}\Delta(P, C)$, where $\zeta(P, C, c_h^*, c_j) = \Delta(P \backslash P_h^*, C \backslash \{c_j\}) - \Delta(P \backslash P_h^*, C)$ is the reassignment cost by swapping $c_j$ out. Otherwise we say that $P_h^*$ is a bad single cluster with $(c_h^*, c_j)$.*

**Definition 2** *Good $t$-Clusters.* *Given an integer $t$ with $t \geq 2$, for a collection of optimal clusters $Q \subseteq \mathbb{P}(C^*)$ with $|Q| \leq t$, $Q$ is called good with a pair of sets $(Z'(Q), V)$ such that $V \subseteq C$, $|V| = |Z'(Q)|$ and $\Delta(Z(Q), C) - \zeta(P, C, Z'(Q), V) - 9\Delta(Z(Q), C^*) > \frac{1}{100k}\Delta(P, C)$, where $\zeta(P, C, Z'(Q), V) = \Delta(P \backslash Z(Q), C \backslash V) - \Delta(P \backslash Z(Q), C)$ is the reassignment cost by swapping the points in $V$ out. Otherwise we say that $Q$ is a set of bad $t$-clusters with $(Z'(Q), V)$.*

---

[2]Note that the authors did not try to optimize the constants

---

**Algorithm 2** MLS

---

**Input**: An instance $(P, k)$ of the $k$-means problem, parameters $T$ and $t$.

**Output**: A set $C \subseteq \mathbb{R}^d$ of centers with size at most $k$.

1: Initialize $C = k$-means++$(P, k)$.
2: **for** $i = 1$ to $T$ **do**
3:    $I = \emptyset$.
4:    **for** $j = 1$ to $t$ **do**
5:        Pick a point $p \in P$ with probability $\Delta(\{p\}, C)/\Delta(P, C)$ and add $p$ to $I$.
6:    **if** $\exists\, U \subseteq I$ and $V \subseteq C$ s.t. $|U| = |V|$ and $\Delta(P, C\backslash V \cup U) < (1 - \frac{1}{100k})\Delta(P, C)$ **then**
7:        $C = C\backslash V \cup U$.
8: **return** $C$.

---

The above definitions estimate the changes of clustering cost by replacing a set of centers $V \subseteq C$ with a set $Q'$ of data points that are close to a subset of optimal clustering centers, where a new clustering is constructed by reassigning data points in $Z(Q)$ to $Q'$ and data points in $P\backslash Z(Q)$ to $C\backslash V$. The main idea behind Algorithm 2 is to iteratively find data points close enough to the optimal clustering centers for swap to make a reduction on clustering cost by at least $(1 - \Theta(\frac{1}{k}))$. In the following, we will prove that with probability at least $\Omega(k^{-t})$, the sampling and swap process induces a significant reduction on clustering cost in each iteration.

We start by dividing the optimal clusters into several groups to give an upper bound of reassignment costs. Given a swap size $t$, for each optimal center $c_h^* \in C^*$, we define $\Psi(c_h^*)$ as a mapping function that maps $c_h^*$ to its closest center in $C$. For simplicity, we say that $c_h^*$ is captured by $\Psi(c_h^*)$. For a center $c_j \in C$, let $\Psi^{-1}(c_j)$ be the set of optimal centers captured by $c_j$. If $|\Psi^{-1}(c_j)| = 0$, then $c_j$ is called a lonely center. If $|\Psi^{-1}(c_j)| = 1$, let $c_h^* = \Psi^{-1}(c_j)$. Then it is called that $(c_h^*, c_j)$ forms a type-1 matched swap pair. If $1 < |\Psi(c_j)| \leq t$, let $\sigma_h$ be an arbitrary set of unused lonely centers with $|\sigma_h| = |\Psi^{-1}(c_j)| - 1$. Let $A = \Psi^{-1}(c_j)$ and $A' = \{c_j\} \cup \sigma_h$. Then, it is called that $(A, A')$ forms a type-1 matched swap set. Let $M_1 = \{(c_h^*, c_j) : (c_h^*, c_j) \text{ is a type-1 matched swap pair}\}$ and $M_2 = \{(A, A') : (A, A') \text{ is a type-1 matched swap set}\}$ be the collections of type-1 matched swap pairs and type-1 matched swap sets, respectively. If $|\Psi^{-1}(c_j)| > t$, then find each lonely center $c_q \in C$ that has not been used for constructing type-1 matched swap pairs or sets. For each lonely center $c_q$ and each $c_h^* \in \Psi^{-1}(c_j)$, $(c_h^*, c_q)$ forms a type-2 matched swap pair. We use $M_3 = \{(c_h^*, c_q) : (c_h^*, c_q) \text{ is a type-2 matched swap pair}\}$ to denote the set of type-2 matched swap pairs. For a set $V \subseteq C$ of centers, let $X(V) = \cup_{c_h \in V} P_h$ be the set of data points in $P$ whose closest centers are in $V$. Given a subset $S \subseteq C$ of clustering centers, we also use $J(S) = \{P_h : c_h \in G\}$ to denote the set of clusters whose centers are in $S$. The following lemma gives upper bounds of reassignment cost by matched swap pairs or sets.

**Lemma 2** *Given a type-1 or type-2 matched swap pair $(c_h^*, c_j)$, it holds that $\zeta(P, C, c_h^*, c_j) \leq 24\Delta(P_j, C^*) + \frac{1}{5}\Delta(P_j, C)$. Given a type-1 matched swap set $(Q, V)$, it holds that $\zeta(P, C, Q, V) \leq 24\Delta(X(V), C^*) + \frac{1}{5}\Delta(X(V), C)$.*

Let $H_1 = \{c_h^* : (c_h^*, c_j) \in M_1\}$ be the set of optimal centers that participate in constructing type-1 matched swap pair. Let $H_2 = \{A : (A, A') \in M_2\}$ be the collection of the subsets of optimal centers that participate in constructing type-1 matched swap set. Let $L = \{c_h^* : (c_h^*, c_q) \in M_3\}$ be the set of optimal centers that participate in constructing type-2 matched swap pair. Let $H_1' = \{c_j : (c_h^*, c_j) \in M_1\}$ be the set of centers in $C$ that participate in constructing type-1 matched swap pair. Let $H_2' = \{A' : (A, A') \in M_2\}$ be the collection of the subsets of centers in $C$ that participate in constructing type-1 matched swap set. Let $L' = \{c_q : (c_h^*, c_q) \in M_3\}$ denote the set of centers in $C$ that participate in constructing type-2 matched swap pair.

During the sampling and swap process in steps 4-7 of Algorithm 2, there are two cases that may happen: (1) $\exists\, c_h^* \in L$ such that $P_h^*$ is a good single cluster with a type-2 matched swap pair $(c_h^*, c_q) \in M_3$; (2) $\forall\, c_h^* \in L$, $P_h^*$ is a bad single cluster with any type-2 matched swap pair $(c_h^*, c_q) \in M_3$. We will discuss the two cases separately in the following. If case (1) happens, we first show that with probability at least $\Omega(\frac{1}{k})$, the clustering cost can be reduced at least by $1 - \Theta(\frac{1}{k})$. Let $c_h^* \in L$ be an optimal center such that $P_h^*$ is a good single cluster with a type-2 matched swap pair $(c_h^*, c_q) \in M_3$. By the definition of good single cluster, we have $\Delta(P_h^*, C) \geq 9\Delta(P_h^*, \{c_h^*\})$.

Define $\chi(P_h^*) = \{p \in P_h^* : d(p, c_h^*) \leq \frac{1.5\Delta(P_h^*, \{c_h^*\})}{|P_h^*|}\}$ as the set of data points in $P_h^*$ that are close to the optimal center $c_h^*$. Observe that $|\chi(P_h^*)| \geq \frac{1}{3}|P_h^*|$. Otherwise, the clustering cost of data points in $P_h^* \backslash \chi(P_h^*)$ is at least $\Delta(P_h^*, \{c_h^*\})$ using $c_h^*$ as center, which contradicts with $\Delta(P_h^* \backslash \chi(P_h^*), \{c_h^*\}) < \Delta(P_h^*, \{c_h^*\})$.

Next, we will show that whenever an optimal cluster $P_h^*$ has large clustering cost with respect to $C$, i.e., $\Delta(P_h^*, C) = b\Delta(P_h^*, \{c_h^*\})$ for a real number $b \geq 3$, it suffices to use data points in $\chi(P_h^*)$ to approximate the clustering cost of $P_h^*$.

**Lemma 3** *Let $P_h^*$ be an optimal cluster with $\Delta(P_h^*, C) = b\Delta(P_h^*, \{c_h^*\})$ for a real number $b \geq 3$. Then, $\Delta(\chi(P_h^*), C) \geq \frac{1}{200}(b-1)\Delta(P_h^*, \{c_h^*\})$.*

We now argue that the clustering cost of each good single cluster takes a certain fraction of the total clustering cost. Then, by sampling according to the squared distances, with good probability, data points close to the center of a good single cluster can be sampled. According to the definition of good single cluster, we can assume that $\Delta(P_h^*, C) = b\Delta(P_h^*, \{c_h^*\})$ for a real number $b \geq 9$. By Lemma 3, we know that $\Delta(\chi(P_h^*), C) \geq \frac{b-1}{200}\Delta(P_h^*, \{c_h^*\}) = \frac{b-1}{200b}\Delta(P_h^*, C) \geq \frac{1}{300}\Delta(P_h^*, C)$. By the definition of good single cluster, we also have $\Delta(P_h^*, C) \geq \frac{1}{100k}\Delta(P, C)$, which implies that $\Delta(\chi(P_h^*), C) \geq \frac{1}{30000k}\Delta(P, C)$. Thus, in each round of the sampling process in steps 4-5 of Algorithm 2, with probability at least $\Omega(\frac{1}{k})$, we can sample a point $q \in \chi(P_h^*)$ for a good single cluster $P_h^*$ with a type-2 matched swap pair $(c_h^*, c_q) \in M_3$. The following lemma shows that, by swapping $q$ with $c_q$ and assigning all the data points in $P_h^*$ to $q$, the clustering cost can be reduced at least by $1 - \Theta(\frac{1}{k})$.

**Lemma 4** *By swapping $q$ with $c_q$, the clustering cost of $\Delta(P, C)$ can be reduced at least by $1 - \Theta(\frac{1}{k})$.*

We have shown that if case (1) happens, we can sample a data point $q$ close to the center of a good single cluster $P_h^*$ to make the clustering cost reduced significantly. Next, we assume that case (1) never happens and case (2) happens. In case (2), the idea behind is to sample data points close to the optimal centers in $H_1$ or sets of optimal centers in $H_2$ to reduce the clustering cost. We first bound the clustering cost of optimal clusters in $J(L)$.

**Lemma 5** *If case (2) happens, then $\Delta(X(L), C) \leq (1 + \frac{1}{t})(9\Delta(X(L), C^*) + 24\Delta(X(L'), C^*) + \frac{1}{5}\Delta(X(L'), C) + \frac{1}{100}\Delta(P, C))$.*

Now, consider a set of centers $Q \in H_2$. We will divide the optimal clusters in $J(Q)$ into two groups. Let $Q_L = \{P_h^* \in J(Q) : \Delta(P_h^*, C) \geq \frac{1}{30000k2^{t-1}}\Delta(P, C)\}$ be the set of optimal clusters in $J(Q)$ with large clustering cost with respect to $C$ and $Q_S = \{P_h^* \in J(Q) : \Delta(P_h^*, C) < \frac{1}{30000k2^{t-1}}\Delta(P, C)\}$ be the set of optimal clusters in $J(Q)$ with small clustering cost with respect to $C$, respectively. We define $Q_S' = \{P_h^* \in Q_S : \Delta(P_h^*, C) < 3\Delta(P_h^*, \{c_h^*\})\}$ as the set of optimal clusters in $Q_S$ whose clustering centers are close to one of the centers in $C$. Let $Q_S'' = Q_S \backslash Q_S'$ and $Q_T = Q_L \cup Q_S''$, respectively. We first show that, it suffices to only consider optimal clusters in $Q_T$.

**Lemma 6** *Let $Q \in H_2$ be a set of centers in $H_2$, where $J(Q)$ is a set of good $t$-clusters with a type-1 matched swap set $(Q, A') \in M_2$. Define $V = A' \backslash \{c_j\}$, where $c_j$ is the center in $A'$ with $|\Psi^{-1}(c_j)| > 1$. Let $U \subseteq P$ be the set of data points with $|U| = |V|$ such that $U \cap \chi(P_h^*) \neq \emptyset$ holds for each $P_h^* \in Q_T$. Then, $\Delta(P, C \backslash V \cup U) \leq (1 - \frac{1}{100k})\Delta(P, C)$.*

Note that there are $t$ sampling iterations in each step 4 of Algorithm 2. Let $H_2^* = \{P_h^* : P_h^* \in Q_T, Q \in H_2\}$ and $H_t = J(H_1) \cup H_2^*$. We will define a mapping function $m$ which maps each $P_h^* \in H_t$ to an integer $m(P_h^*) \in [t]$. For each $P_h^* \in H_t$ such that $c_h^* \in H_1$, we define $m(P_h^*) = 1$. Then, consider each $Q \in H_2$. For each $P_h^* \in Q_T$, we define $m(P_h^*) = i$ such that $i \in [|Q_T|]$ and $m(P_h^*) \neq m(P_j^*)$ for any two optimal clusters $P_h^*, P_j^* \in Q_T$. For each $Q \in H_2$, let $p_s(Q)$ be the success probability that for each $P_h^* \in Q_T$, a data point is sampled from $\chi(P_h^*)$ in the $m(P_h^*)$-th iteration of Algorithm 2. For each $c_h^* \in H_1$, let $p_s(c_h^*)$ be the success probability that a data point is sampled from $\chi(P_h^*)$ in the first iteration (note that in this case $m(P_h^*) = 1$) of step 4. Let $H_G^1 = \{c_h^* \in H_1 : P_h^*$ is a good single cluster with the type-1 matched swap pair $(c_h^*, c_j) \in M_1\}$ be the set of centers in $H_1$ whose optimal clusters are good single clusters with type-1 matched swap pairs in $M_1$. Let $H_G^2 = \{Q \in H_2 : J(Q)$ is a set of good $t$-clusters with the type-1 matched swap set $(Q, V) \in M_2\}$ be the collection of the sets of centers in $H_2$ whose corresponding optimal clusters are good

$t$-clusters with type-1 matched swap sets in $M_2$. Define $p_s^f = \sum_{c_h^* \in H_G^1} p_s(c_h^*) + \sum_{Q \in H_G^2} p_s(Q)$ as the summation of the success probability. Since all the optimal clusters in $J(L)$ belong to bad single cluster, and the events defined related to $p_s(Q)$ or $p_s(c_h^*)$ for each $Q \in H_2$ and $c_h^* \in H_1$ are mutually exclusive, $p_s^f$ gives a lower bound success probability to get a $(1 - \frac{1}{100k})$ reduction on the clustering cost in each iteration of steps 2-7. In the following, we will show how to obtain a lower bound for $p_s^f$.

Consider a set $Q \in H_2$ of optimal centers such that $J(Q)$ is a set of good $t$-clusters with a type-1 matched swap set $(Q, A')$. There are two subcases that may happen: (1) $\exists P_h^* \in J(Q)$ such that $P_h^*$ is a good single cluster with a swap pair $(c_h^*, c_j)$, where $c_j$ is a lonely center used for constructing type-1 matched swap set; (2) $\forall P_h^* \in J(Q)$, $P_h^*$ is a bad single cluster with any swap pair $(c_h^*, c_j)$, where $c_j$ is a lonely center used to construct type-1 matched swap set. In subcase (1), since $t$ is usually a constant and could be much smaller than $k$, with probability at least $\Omega(k^{-1})$, a data point $q \in \chi(P_h^*)$ can be sampled in the first iteration of step 4 in Algorithm 2 for swap to make the clustering cost reduced at least by $(1 - \frac{1}{100k})$ according to Lemma 4. Next, we assume that subcase (1) never happens and subcase (2) happens. In subcase (2), our objective is to sample a set $V$ of data points such that $V$ contains at least one point from $\chi(P_h^*)$ for each $P_h^* \in Q_T$. For each optimal cluster $P_h^* \in Q_L$, with probability at least $\Omega(k^{-1})$, we can sample a data point $q \in \chi(P_h^*)$ in the $m(P_h^*)$-th iteration of step 4 in Algorithm 2. Thus, the probability can be bounded by $\Omega(k^{-|Q_L|})$. Then, we only need to consider optimal clusters in $Q_S$. By Lemma 6, it suffices to consider optimal clusters in $Q_S''$. The following lemma gives an upper bound of the failure probability of not sampling a data point from $\chi(P_h^*)$ for each $P_h^* \in Q_S''$ in the $m(P_h^*)$-th iteration in step 4 of Algorithm 2.

**Lemma 7** *Given a set $Q \in H_2$ of centers such that $J(Q)$ is a set of good $t$-clusters, the probability that step 4 of Algorithm 2 fails to sample a data point $q$ from $\chi(P_h^*)$ for each $P_h^* \in Q_S''$ in the $m(P_h^*)$-th iteration is at most $(1 + \frac{1}{30000k})e^{-\Delta(Z(Q_S''),C)/(300\Delta(P,C))}$.*

Now we can bound the success probabilities of $p_s(Q)$ and $p_s(c_h^*)$ for each $Q \in H_2$ and $c_h^* \in H_1$. We will divide the optimal clusters in $\mathbb{P}(C^*) \backslash J(L)$ into two different groups $H_G$ and $H_B$. Firstly, we consider the centers in $H_1$. For a center $c_h^* \in H_1$, if $P_h^*$ is a good single cluster with the type-1 matched swap pair $(c_h^*, c_j) \in M_1$, then add $P_h^*$ to $H_G$. Otherwise add $P_h^*$ to $H_B$. For a set $Q \in H_2$ of optimal clustering centers, if $J(Q)$ is a set of good $t$-clusters with the type-1 matched swap set $(Q, V) \in M_2$, then add each $P_h^* \in Q_S''$ to $H_G$ and each $P_h^* \in J(Q) \backslash Q_S''$ to $H_B$. If $J(Q)$ is a set of bad $t$-clusters with the type-1 matched swap set $(Q, V) \in M_2$, then add each $P_h^* \in J(Q)$ to $H_B$. The following lemma argues that the summation clustering cost of the optimal clusters in $H_G$ is large.

**Lemma 8** *For the optimal clusters in $H_G$, we have $\Delta(Z(H_G), C) \geq \frac{1}{100}\Delta(P, C)$.*

For a good single cluster $P_h^*$ with the type-1 matched swap pair $(c_h^*, c_j) \in M_1$ where $c_h^* \in H_1$, define $p_f(c_h^*) = 1 - p_s(c_h^*)$ as the probability that the first iteration in step 4 of Algorithm 2 fails to sample a data point $q \in \chi(P_h^*)$. Then, we have $p_f(c_h^*) \leq e^{-p_s(c_h^*)} \leq e^{-\Delta(P_h^*,C)/300\Delta(P,C)}$ by Lemma 3. For a set $Q \in H_G^2$ of optimal centers, we have that $p_f(Q_S'')$ is the failure probability of not sampling a data point from $\chi(P_h^*)$ for each $P_h^* \in Q_S''$ in the $m(P_h^*)$-th iteration of step 4 in Algorithm 2. Recall that $p_s^f = \sum_{c_h^* \in H_G^1} p_s(c_h^*) + \sum_{Q \in H_2} p_s(Q)$ is the summation of success probability. Then, we have $p_s^f \geq \Omega(k^{-t})(\sum_{c_h^* \in H_G^1} p_s(c_h^*) + \sum_{Q \in H_G^2} p_s(Q_S''))$. Let $p_s^{f'} = \sum_{c_h^* \in H_G^1} p_s(c_h^*) + \sum_{Q \in H_G^2} p_s(Q_S'')$. Observe that $p_s^{f'} \geq 1 - \prod_{c_h^* \in H_G^1}(1 - p_s(c_h^*))\prod_{Q \in H_G^2}(1 - p_s(Q_S''))$. Since there are at most $\frac{k}{2}$ good $t$-clusters, by Lemma 7, we have $p_s^{f'} \geq 1 - (1 + \frac{1}{30000k})^{\frac{k}{2}} e^{-\frac{\Delta(Z(H_G),C)}{300\Delta(P,C)}} \geq 1 - e^{\frac{k}{2}\ln(1+\frac{1}{30000k}) - \frac{\Delta(Z(H_G),C)}{300\Delta(P,C)}} \geq 1 - e^{\frac{1}{60000} - \frac{1}{30000}} \geq 1 - e^{-\frac{1}{60000}} \geq \frac{1}{60001}$, where the third inequality follows from Lemma 8 and $\ln(1 + x) \leq x$. Then, it holds that $p_s^f \geq \Omega(k^{-t})p_s^{f'} = \Omega(k^{-t})$, which indicates that with probability at least $\Omega(k^{-t})$, we can sample data points close to a set of good $t$-clusters or a good single cluster for swap to make the clustering cost reduced at least by $(1 - \frac{1}{100k})$ according to Lemma 6. Putting all things together, Theorem 2 can be proved (Detailed proof of Theorem 2 is given in Appendix A).

**Running Time Analysis.** By Theorem 2, in order to obtain a $(50(1 + \frac{1}{t}) + \epsilon)$-approximate solution, the iteration rounds for Algorithm 2 should be $O(k^{t+1}\log(\epsilon^{-1}\log k))$. In each iteration, it takes $O(ndk)$ time to update the distances between data points to their closest centers. During the sampling process, $t$ data points are sampled according to the $D^2$-Sampling distribution to serve as the candidate set of centers for swapping in, which takes time $O(nt)$ if the distances from data points to their

centers are already known. It takes $O(k^t)$ time to enumerate each subset with size at most $t$ of the set of current centers opened. It takes $O(ndk^t)$ time to recalculate the clustering cost after each swap if $t$ nearest centers of each data point are maintained during the whole process. Thus, the total running time of Algorithm 2 is $O(ndk^{2t+1} \log(\epsilon^{-1} \log k))$.

## 3.2 Accelerating Multi-Swap Local Search for $k$-means

In this subsection, we provide a more practical algorithm for the $k$-means problem to accelerate the proposed multi-swap local search process. The algorithm is given in Algorithm 3. The main idea behind is to use sampling-based methods to obtain fast clustering cost updating during swaps. In step 7 of Algorithm 3, a small sample set $S$ of size $\frac{2k}{\epsilon} \log \frac{k}{\eta}$ is randomly taken from $P$. For clustering cost updating, instead of calculating the clustering cost of all the data points in $P$, we use the clustering cost of $S \subseteq P$ as an estimation. This reduces the time for picking the best swap pair during a single local search iteration from $O(nd)$ to $O(poly(k)d)$. Then, in steps 16-18, we design a recombination method to find better initialization which prevents the local search algorithm from falling into a poor local optimum too early. In step 16 of Algorithm 3, we randomly take a set $D \subseteq P$ of centers with size $O(k \log k)$. Let $C_1 = C \cup D$ be the set of the new candidate centers. For each center $c_h \in C_1$, we add a score of $\Delta(P_h, C)/\Delta(P, C)$ to it. Similarly, for each center $c_h \in C$, we also add a score $\Delta(P_h, C)/\Delta(P, C)$ to it. Then, by giving each center in $C_1$ and $C$ a score weight of 0.75 and 0.25, respectively, we pick the top-$k$ data points with the highest scores as a new initialization of clustering centers to find potentially better clustering costs until convergence.

---

**Algorithm 3** MLSP

---

**Input**: An instance $(P, k)$ of the $k$-means problem, parameters $T$, $t$, $R'$, $\epsilon$ and $\eta$.
**Output**: A set $C \subseteq \mathbb{R}^d$ of centers with size at most $k$.
 1: Initialize $C = k\text{-means++}(P, k)$, $r = 0$, $C_f = \emptyset$.
 2: **while** $r < R'$ **do**
 3:     **for** $i = 1$ to $T$ **do**
 4:         $I = \emptyset$.
 5:         **for** $j = 1$ to $t$ **do**
 6:             Pick a point $p \in P$ with probability $\Delta(\{p\}, C)/\Delta(P, C)$, and add $p$ to $I$.
 7:         Randomly sample a set $S$ from $P$ of size $\frac{2k}{\epsilon} \log \frac{k}{\eta}$.
 8:         Let $(U, V)$ be a swap set such that $U \subseteq I$, $V \subseteq C$, $|U| = |V|$ and $\Delta(S, C \backslash V \cup U)$ is minimized.
 9:         **if** $\Delta(P, C \backslash V \cup U) < (1 - \frac{1}{100k})\Delta(P, C)$ **then**
10:             $C = C \backslash V \cup U$.
11:     For each center $c \in C$, find the 50-nearest neighbors in $P$ to $C$ for improvements on clustering cost by swapping $c$ with one neighbor until convergence.
12:     **if** $\Delta(P, C) < \Delta(P, C_f)$ **then**
13:         $C_f = C$.
14:     **else**
15:         $r = r + 1$, randomly sample a set $D$ from $P$ with size $\frac{k}{\epsilon} \log \frac{k}{\eta}$, and set $C_1 = C \cup D$.
16:         For each $c_h \in C$, calculate $S'(c_h) = \frac{\Delta(P_h, C)}{\Delta(P, C)}$, and add a score of $0.25 S'(c_h)$ to $c_h$.
17:         For each $c_h \in C_1$, calculate $S'(c_h) = \frac{\Delta(P_h, C)}{\Delta(P, C)}$, and add a score of $0.75 S'(c_h)$ to $c_h$.
18:         Reset $C$ as data points in $C_1$ with top $k$ scores.
19: **return** $C_f$.

---

# 4 Experiments

In this section, we compare our proposed algorithms with the branch and bound solver and other local search methods. For hardware, all the experiments are conducted on 72 Intel Xeon Gold 6230 CPUs with 500GB memory.

**Datasets** We evaluate the performance of our algorithms on 8 datasets used in [15] with sizes over 50,000, two datasets SUSY (5,000,000 × 17) and HIGGS (11,000,000 × 27) from the UCI Machine

Learning Repository[3], and one dataset SIFT ($100,000,000 \times 128$) with size $100,000,000$ used in [14]. In Appendix B, we give experiments on small datasets with sizes smaller than $50,000$ used in [15].

**Experimental Setup** In our experiments, we choose centers from the datasets such that the problem can be solved by branch and bound solver (denoted as BB for short), which can serve as references of optimal solutions. We compare our MLSP and MLS algorithms with the BB method and other local search algorithms. Following the settings in [15], the centers obtained by different algorithms are projected to their closest data points in $P$, and the number of clusters $k$ is set to be $3, 5$ and $10$. Each algorithm is executed for 10 times, and the average results with deviation, the best results and the average running time are given. We test the performance of different choices of parameters in Appendix B. We also conduct the experiments with fixed running time in Appendix B.

**Algorithms** In our experiments, we consider six algorithms. The first is the BB method in [15], which is the state-of-the-art solver for handling large-scale datasets. For local search algorithms, we compare our MLS and MLSP algorithms with the LS++ algorithm in [11] and FLS algorithm in [9], using the Lloyd's algorithm [12] as a baseline. For MLS algorithm, we use the sampling-based method designed for MLSP to accelerate the swapping process. For MLSP and MLS algorithms, we set the sampling rounds as $T = 400$ with a swap size $t = 2$. For MLSP algorithm, the failure upper bound is set to be $R' = 5$. For parameters $\epsilon$ and $\eta$, we fix them as $0.5$. For fair comparison, we also set the number of sampling rounds as 400 for LS++. Following the settings in [11], for all local search algorithms, the Lloyd's algorithm [12] is used as the final step to adjust the centers.

| Method | Dataset | SampleSize | BB(Cost) | Best Cost | Average Cost | Time(s) | Dataset | SampleSize | BB(Cost) | Best Cost | Average Cost | Time(s) |
|---|---|---|---|---|---|---|---|---|---|---|---|---|
| LS++ | | | | 132.75 | 138.66±4.02 | 6.64 | | | | 2.5720E+06 | 2.5801E+06±7.2E+03 | 55.28 |
| FLS | | | | **131.74** | 136.19±2.78 | 27.25 | | | | 2.5687E+06 | 2.5720E+06±1.8E+03 | 609.22 |
| Lloyd | rds | 50,000*3 | 132.73 | 132.73 | 138.01±3.96 | **0.07** | SPNET_3D | 434,874*3 | OOM | 2.5702E+06 | 2.5816E+06±8.0E+03 | **0.62** |
| MLSP | | | | **131.74** | **132.03±0.45** | 58.61 | | | | **2.5677E+06** | **2.5688E+06±4.7E+02** | 561.06 |
| MLS | | | | 131.78 | 136.71±2.76 | 5.12 | | | | 2.5696E+06 | 2.5778E+06±7.3E+03 | 25.39 |
| LS++ | | | | 6.1546E+07 | 6.2916E+07±1.5E+06 | **4.39** | | | | **560502.36** | 561679.14±1177.73 | 123.86 |
| FLS | | | | **6.1534E+07** | 6.5554E+07±1.1E+06 | 20.99 | | | | 562756.29 | 561856.27±1034.52 | 1087.1 |
| Lloyd | KEGG | 53,413*23 | 6.1564E+07 | 6.1564E+07 | 6.3870E+07±2.3E+06 | **0.21** | syn | 1,000,000*2 | OOM | 562761.91 | 571251.23±796.86 | **0.89** |
| MLSP | | | | **6.1534E+07** | **6.1546E+07±3.7E+04** | 59.71 | | | | **560502.36** | **560737.88±706.23** | 789.83 |
| MLS | | | | 6.1546E+07 | 6.3076E+07±1.9E+06 | 5.68 | | | | **560502.36** | 561444.42±153.94 | **60.61** |
| LS++ | | | | 24787 | 25209±575 | 10.39 | | | | 2.7091E+08 | 2.7302E+08±2.9E+06 | 817.52 |
| FLS | | | | 24786 | 24826±328 | 54.12 | | | | 2.7082E+08 | 2.7083E+08±9.4E+03 | 7148.45 |
| Lloyd | Urban_10 | 100,000*2 | 25123 | 24663 | 25484±696 | **0.09** | USC_1990 | 2,458,685*68 | OOM | 2.7100E+08 | 2.8497E+08±9.5E+06 | **40.32** |
| MLSP | | | | **24659** | **24716±58** | 71.91 | | | | **2.7073E+08** | **2.7079E+08±7.8E+02** | 5687.57 |
| MLS | | | | **24659** | 24889±257 | 5.96 | | | | 2.7082E+08 | 2.7082E+08±2.4E+03 | 619.16 |
| LS++ | | | | 1.3738E+14 | 1.3917E+14±2.2E+12 | 28.13 | | | | 3.2738E+07 | 3.2875E+07±1.1E+05 | 827.21 |
| FLS | | | | **1.3663E+14** | 1.3745E+14±8.0E+11 | 206.67 | | | | 3.1632E+07 | 3.1672E+07±2.8E+05 | 9287.26 |
| Lloyd | RNG_AGR | 199,843*7 | 1.3678E+14 | 1.3671E+14 | 1.3799E+14±2.7E+12 | **0.47** | SUSY | 5,000,000*17 | OOM | 3.1639E+07 | 3.1668E+07±3.1E+05 | **31.87** |
| MLSP | | | | **1.3663E+14** | **1.3685E+14±2.0E+11** | 201.21 | | | | **3.1575E+07** | **3.1633E+07±3.8E+04** | 7462.57 |
| MLS | | | | 1.3701E+14 | 1.3906E+14±1.8E+12 | 14.24 | | | | 3.2219E+07 | 3.2424E+07±1.4E+05 | **534.11** |
| LS++ | | | | 88720 | 89677±1370 | 30.78 | | | | 1.8604E+08 | 1.8834E+08±1.4E+06 | 2424.97 |
| FLS | | | | **88329** | 89233±890 | 183.4 | | | | 1.8938E+08 | 1.8964E+08±1.5E+05 | 39826.29 |
| Lloyd | Urban_GB | 360,177*2 | OOM | 88346 | 92595±4131 | **0.41** | HIGGS | 11,000,000*27 | OOM | 1.8461E+08 | 1.8568E+08±1.2E+06 | **171.06** |
| MLSP | | | | **88329** | **88935±668** | 175.75 | | | | **1.8373E+08** | **1.8410E+08±1.1E+05** | 21928.97 |
| MLS | | | | **88329** | 89557±1333 | **13.07** | | | | 1.8623E+08 | 1.8686E+08±5.5E+05 | **2037.68** |
| LS++ | | | | 1.3994E+13 | 1.4014E+13±8.2E+10 | 121260 | | | | | | |
| FLS | | | | 1.3905E+13 | 1.3985E+13±5.6E+10 | 140560 | | | | | | |
| Lloyd | SIFT | 100,000,000*128 | OOM | 1.3718E+13 | 1.3803E+13±9.6E+10 | **566.99** | | | | | | |
| MLSP | | | | - | - | >48h | | | | | | |
| MLS | | | | **1.3715E+13** | **1.3883E+13±9.0E+10** | 15179 | | | | | | |

Table 2: Comparison results on clustering costs and running time with $k = 10$ on datasets with sizes larger than 50,000, where OOM is short for out of memory

**Results** Table 2 shows the results on 11 datasets with sizes over 50,000 using $k = 10$. For clustering cost, on each dataset, the best clustering cost returned by our MLSP algorithm is smaller than BB method and other local search methods. For each dataset except for RNG_AGR, the average clustering cost returned by our MLSP algorithm outperforms the result of BB method. It can be seen that, as the sizes of datasets grow, BB method requires larger space complexity. When the size of dataset is over 360,000, it requires a memory of over 500GB, which is not practical for handling large-scale datasets. For LS++ method, it is more difficult to find high-quality solutions as the sizes of datasets grow. As for FLS, although the performance of FLS is better than that of LS++, our MLSP algorithm improves the performance of FLS on clustering cost with smaller deviation than FLS. By calculating the average values over all datasets, for MLSP algorithm, we can get that the clustering cost is reduced by 1.9%, 1.3% and 2.4% compared with LS++, FLS and Lloyd's algorithm, respectively. For MLS algorithm, on average, the clustering cost is reduced by 0.6% and 1.1% compared with LS++ and Lloyd's algorithm, respectively. As for running time, it can be seen that our proposed algorithm scales well as the sizes of datasets grow. By calculating the average values over all datasets, our proposed MLS algorithm is at least 1.79 times faster than LS++ algorithm.

---

[3]https://archive.ics.uci.edu/ml/index.php

The experimental results on the performances with varying $T$ and $R'$ (Appendix B.1) show that larger $T$ and $R'$ will not influence the results too much. In general, larger sampling rounds and larger failure upper bound can result in potentially better solutions with higher running time. For parameters $\epsilon$ and $\eta$, the results (Appendix B.1) show that the performances of our proposed algorithms are almost the same for different choices of $\epsilon$ and $\eta$. Tables 3, 4 and 5 show the comparison results of different algorithms with varying iteration rounds on dataset rds, KEGG and Urban_10, respectively. The results show that, our proposed MLSP algorithm always achieves the best clustering cost compared with LS++ and MLS algorithms. Tables 6, 7, and 8 show the results of MLSP algorithm with varying number of failure upper bound $R'$ for fixed $\epsilon = 0.5$, $\eta = 0.5$ and $T = 400$. It can be seen that a larger failure upper bound will lead to smaller deviation, and the running time becomes higher. Tables 9, 10, and 11 show the results of MLSP algorithm with varying parameters $\epsilon$ and $\eta$ for fixed $T = 400$ and $R' = 5$ on dataset rds, KEGG, and Urban_10, respectively. It can be seen that smaller values of $\epsilon$ and $\eta$ result in better performances on clustering cost with smaller deviation, and the running time becomes higher. The experimental results on small datasets (Appendix B.2) suggest that the proposed MLSP method not only outperforms other algorithms in terms of clustering quality but also runs much faster than the BB solver. The experimental results on the performances with different values of $k$ (Appendix B.3) show that our proposed MLSP algorithm can still achieve the best clustering quality for smaller values of $k$. The experimental results on the performances with fixed time limit (Appendix B.4) show that our proposed MLSP algorithm achieves the best clustering quality within any given time constraints.

Tables 12 and 13 show the results on 18 datasets with sizes smaller than $50,000$ using $k = 10$. It can be seen that, for each dataset, the best clustering cost returned by our MLSP algorithm is smaller than BB method and other local search methods. For each dataset, the average clustering cost returned by our MLSP algorithm nearly matches the result of the BB method. As for FLS, although the performance of FLS is better than that of LS++, our proposed MLSP algorithm improves the performance of FLS on clustering cost with smaller deviation on most datasets. As for running time, there is no significant difference on running time among different local search algorithms for small datasets. Tables 14 and 15 present the performances of different local search algorithms with $k = 3$ on small datasets. Table 18 presents the performances of different local search algorithms with $k = 3$ on large datasets. Tables 16 and 17 present the performances of different local search algorithms with $k = 5$ on small datasets. Table 19 presents the performances of different local search algorithms with $k = 5$ on large datasets. It can be seen that, our proposed MLSP algorithm achieves the best clustering performance on most datasets with different values of $k$. On each dataset, the best clustering cost returned by our MLSP algorithm matches the clustering cost of BB method. On each dataset, the average clustering cost returned by our MLSP algorithm nearly matches the result of the BB method. As for running time, there is no significant difference on running time among all local search algorithms on small datasets. However, as the data sizes grow, our proposed MLS algorithm becomes much faster than other local search algorithms.

## 5   Conclusion

In this paper, we propose fast local search algorithms for the $k$-means problem with multi-swap strategy, which runs in linear time in the data size. We develop new sampling techniques, which accelerate the process of clustering cost update during swaps. By proposing a recombination mechanism, the proposed algorithm can find potentially better solutions. Experimental results show that our algorithms achieve better performance on both small and large datasets compared with the state-of-the-art algorithms. An interesting future direction is how to design fast local search approximation algorithms for handling high dimensional clustering datasets.

## Acknowledgments

This work was supported by National Natural Science Foundation of China (62172446, 62350004, 62332020), Open Project of Xiangjiang Laboratory (22XJ02002, 22XJ03005), and Central South University Research Programme of Advanced Interdisciplinary Studies (2023QYJC023). This work was also carried out in part using computing resources at the High Performance Computing Center of Central South University.

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

## A   Missing Proofs in Section 3

**Lemma 2.** *Given a type-1 or type-2 matched swap pair $(c_h^*, c_j)$, it holds that $\zeta(P, C, c_h^*, c_j) \le 24\Delta(P_j, C^*) + \frac{1}{5}\Delta(P_j, C)$. Given a type-1 matched swap set $(Q, V)$, it holds that $\zeta(P, C, Q, V) \le 24\Delta(X(V), C^*) + \frac{1}{5}\Delta(X(V), C)$.*

**Proof** We first consider a type-1 matched swap set $(Q, V)$. For a data point $p \in P$, we use $s_p$ to denote its closest center in $C$. Let $o_p$ be its closest optimal center in $C^*$. Observe that

$$\zeta(P, C, Q, V) \le \sum_{p \in X(V) \backslash Z(J(Q))} d(p, s_{o_p}) - d(p, s_p)$$

$$\le \sum_{p \in X(V) \backslash Z(J(Q))} \left( \sqrt{d(p, o_p)} + \sqrt{d(o_p, s_{o_p})} \right)^2 - d(p, s_p)$$

$$\le \sum_{p \in X(V) \backslash Z(J(Q))} \left( \sqrt{d(p, o_p)} + \sqrt{d(o_p, s_p)} \right)^2 - d(p, s_p)$$

$$\le \sum_{p \in X(V) \backslash Z(J(Q))} \left( 2\sqrt{d(p, o_p)} + \sqrt{d(p, s_p)} \right)^2 - d(p, s_p)$$

$$\le \sum_{p \in X(V) \backslash Z(J(Q))} 4d(p, o_p) + 2\sqrt{\frac{2}{\lambda}}\sqrt{2\lambda d(p, o_p)d(p, s_p)}$$

$$\le \sum_{p \in X(V) \backslash Z(J(Q))} (4 + \frac{2}{\lambda})d(p, o_p) + 2\lambda d(p, s_p),$$

where the first inequality follows from the fact that $s_{o_p}$ is still in $C\backslash V$ for each data point $p \in X(V)\backslash Z(J(Q))$ according to the definition of the swap set defined, the second and fourth steps follow from the triangle inequality, the third step follows from the fact that $s_{o_p}$ is the nearest point to $o_p$ in $C$, and the last step follows from Cauchy Inequality. Let $\lambda = \frac{1}{10}$. Then $\zeta(P, C, Q, V) \le 24\Delta(X(V), C^*) + \frac{1}{5}\Delta(X(V), C)$. The cases for matched swap pair are similar to the cases for matched swap set. For a matched swap pair $(c_h^*, c_j)$, we can get that $\zeta(P, C, c_h^*, c_j) \le 24\Delta(P_j, C^*) + \frac{1}{5}\Delta(P_j, C)$. $\qquad\square$

**Lemma 3.** *Let $P_h^*$ be an optimal cluster with $\Delta(P_h^*, C) = b\Delta(P_h^*, \{c_h^*\})$ for a real number $b \ge 3$. Then, $\Delta(\chi(P_h^*), C) \ge \frac{1}{200}(b-1)\Delta(P_h^*, \{c_h^*\})$.*

**Proof** Let $s_{c_h^*}$ be the closest center in $C$ to $c_h^*$. Observe that $d(c_h^*, s_{c_h^*}) \ge \frac{(b-1)\Delta(P_h^*, \{c_h^*\})}{|P_h^*|}$. Otherwise $\Delta(P_h^*, \{s_{c_h^*}\}) < b\Delta(P_h^*, \{c_h^*\})$ holds according to Lemma 1. Consider an arbitrary data point $p \in \chi(P_h^*)$. By triangle inequality, we have

$$\sqrt{\Delta(\{p\}, C)} \ge \sqrt{d(c_h^*, s_{c_h^*})} - \sqrt{d(p, c_h^*)} \ge \sqrt{\frac{\Delta(P_h^*, \{c_h^*\})}{|P_h^*|}}(\sqrt{b-1} - \sqrt{1.5}).$$

Since $\sqrt{b-1} \ge \sqrt{2}$, we have

$$\Delta(\{p\}, C) \ge \left( 1 - \sqrt{\frac{3}{4}} \right)^2 \frac{(b-1)\Delta(P_h^*, \{c_h^*\})}{|P_h^*|}.$$

Together with the fact that $|\chi(P_h^*)| \ge \frac{1}{3}|P_h^*|$, we have $\Delta(\chi(P_h^*), C) \ge \frac{b-1}{200}\Delta(P_h^*, \{c_h^*\})$, which proves the Lemma. $\qquad\square$

**Lemma 4.** *By swapping $q$ with $c_q$, the clustering cost of $\Delta(P, C)$ can be reduced at least by $1 - \Theta(\frac{1}{k})$.*

**Proof** Observe that $\Delta(P_h^*, \{q\}) < 9\Delta(P_h^*, \{c_h^*\})$ holds by Lemma 1. Then, we can get that

$$
\begin{aligned}
\Delta(P, C\backslash\{c_q\} \cup \{q\}) &= \Delta(P, C) - (\Delta(P, C) - \Delta(P, C\backslash\{c_q\} \cup \{q\})) \\
&= \Delta(P, C) - (\Delta(P_h^*, C) + \Delta(P\backslash P_h^*, C) - \Delta(P_h^*, C\backslash\{c_q\} \cup \{q\}) \\
&\quad - \Delta(P\backslash P_h^*, C\backslash\{c_q\} \cup \{q\})) \\
&\leq \Delta(P, C) - (\Delta(P_h^*, C) - \zeta(P, C, c_h^*, c_q) - 9\Delta(P_h^*, \{c_h^*\})) \\
&\leq (1 - \frac{1}{100k})\Delta(P, C),
\end{aligned}
$$

where the second to the last inequality follows from the fact that $\Delta(P_h^*, \{q\}) < 9\Delta(P_h^*, \{c_h^*\})$, and the last inequality follows from the definition of good single cluster. $\qquad\square$

**Lemma 5.** *If case (2) happens, then $\Delta(X(L), C) \leq (1 + \frac{1}{t})(9\Delta(X(L), C^*) + 24\Delta(X(L'), C^*) + \frac{1}{5}\Delta(X(L'), C) + \frac{1}{100}\Delta(P, C))$.*

**Proof** By the condition of case (2), it holds that all the optimal clusters in $J(L)$ belong to bad single cluster. For any bad single cluster $P_h^* \in J(L)$, let $(c_h^*, c_h^m) \in M_3$ be the type-2 matched swap pair with minimum reassignment cost, i.e., $(c_h^*, c_h^m) = \arg\min_{(c_h^*, c_n) \in M_3} \zeta(P, C, c_h^*, c_n)$. According to the definition of bad single cluster, we have

$$
\Delta(P_h^*, C) \leq \frac{1}{100k}\Delta(P, C) + 9\Delta(P_h^*, \{c_h^*\}) + \zeta(P, C, c_h^*, c_h^m).
$$

Let $\kappa(M_3)$ be the set of lonely centers used for constructing type-2 matched swap pair. For an optimal center $c_h^* \in C^*$, denote $s_{c_h^*}$ as the center in $C$ closest to $c_h^*$. Let $L_1 = \cup_{c_h^* \in L} s_{c_h^*}$. By the definitions of $\kappa(M_3)$ and $L$, for each $s_{c_h^*} \in L_1$, we can find a set $z(s_{c_h^*}) \subseteq \kappa(M_3)$ with size $|\Psi^{-1}(s_{c_h^*})| - 1$ such that $z(a) \cap z(b) = \emptyset$ for any $a, b \in L_1$. For each $s_{c_h^*} \in L_1$, since $|\Psi^{-1}(s_{c_h^*})| > t$, it holds that $|\Psi^{-1}(s_{c_h^*})|/|z(s_{c_h^*})| \leq 1 + \frac{1}{t}$. By taking a summation over all the centers in $L$, we have $|L| = \sum_{c_h^* \in L} 1 = \sum_{s_{c_h^*} \in L_1} |\Psi^{-1}(s_{c_h^*})| \leq \sum_{s_{c_h^*} \in L_1} |z(s_{c_h^*})|(1 + \frac{1}{t}) \leq |\kappa(M_3)|(1 + \frac{1}{t})$. Then, by considering all the optimal centers in $L$ and taking a summation, we have

$$
\begin{aligned}
\Delta(X(L), C) = \sum_{c_h^* \in L} \Delta(P_h^*, C) &\leq \sum_{c_h^* \in L} \frac{1}{100k}\Delta(P, C) + 9\Delta(P_h^*, \{c_h^*\}) + \zeta(P, C, c_h^*, c_h^m) \\
&\leq \frac{1}{100}\Delta(P, C) + 9\Delta(X(L), C^*) + \sum_{c_h^* \in L} \frac{\sum_{c \in \kappa(M_3)} \zeta(P, C, c_h^*, c)}{|\kappa(M_3)|} \\
&\leq (1 + \frac{1}{t})(9\Delta(Z(J(L)), C^*) + 24\Delta(X(L'), C^*) \\
&\quad + \frac{1}{5}\Delta(X(L'), C) + \frac{1}{100}\Delta(P, C)),
\end{aligned}
$$

where the second to the last inequality follows from the fact that $c_h^m$ is the center with minimum reassignment cost, and the last inequality follows from Lemma 2 and the fact that $|L|/|\kappa(M_3)| \leq 1 + t^{-1}$. $\qquad\square$

**Lemma 6.** *Let $Q \in H_2$ be a set of centers in $H_2$, where $J(Q)$ is a set of good t-clusters with a type-1 matched swap set $(Q, A') \in M_2$. Define $V = A'\backslash\{c_j\}$, where $c_j$ is the center in $A'$ with $|\Psi^{-1}(c_j)| > 1$. Let $U \subseteq P$ be the set of data points with $|U| = |V|$ such that $U \cap \chi(P_h^*) \neq \emptyset$ holds for each $P_h^* \in Q_T$. Then, $\Delta(P, C\backslash V \cup U) \leq (1 - \frac{1}{100k})\Delta(P, C)$.*

**Proof** Consider an arbitrary optimal cluster $P_h^* \in Q_S'$. We know that $\Delta(P_h^*, C) < 3\Delta(P_h^*, \{c_h^*\})$ by the definition of $Q_S'$. Let $s_{c_h^*}$ be the closest center in $C$ to $c_h^*$. We first argue that $s_{c_h^*}$ is close to $c_h^*$ with $d(c_h^*, s_{c_h^*}) \leq \frac{8\Delta(P_h^*, \{c_h^*\})}{|P_h^*|}$. For a data point $p \in P_h^*$, let $s_p$ be its closest center in $C$. According

to the triangle inequality, we have

$$d(c_h^*, s_{c_h^*}) \leq \frac{\sum_{p \in P_h^*} d(c_h^*, s_p)}{|P_h^*|} \leq \frac{1}{|P_h^*|} \sum_{p \in P_h^*} (1 + \lambda)d(c_h^*, p) + (1 + \frac{1}{\lambda})d(p, s_p)$$

$$\leq \frac{(1 + \lambda)\Delta(P_h^*, \{c_h^*\}) + (1 + \frac{1}{\lambda})\Delta(P_h^*, C)}{|P_h^*|}$$

$$\leq \frac{8\Delta(P_h^*, \{c_h^*\})}{|P_h^*|},$$

where the last inequality follows from $\Delta(P_h^*, C) < 3\Delta(P_h^*, \{c_h^*\})$ and feeding $\lambda = \sqrt{3}$. By assigning all the data points in $P_h^*$ to $s_{c_h^*}$, we have $\Delta(P_h^*, \{s_{c_h^*}\}) \leq 9\Delta(P_h^*, \{c_h^*\})$ by Lemma 1. Then, by swapping $U$ with $V$, we have that

$$\Delta(P, C\backslash V \cup U) = \Delta(P, C) - (\Delta(P, C) - \Delta(P, C\backslash V \cup U))$$

$$\leq \Delta(P, C) - (\Delta(Z(J(Q)), C) + \Delta(P\backslash Z(J(Q)), C) - \Delta(Z(Q_T), U)$$

$$- \Delta(Z(Q_S'), \{s_{c_h^*}\}) - \Delta(P\backslash Z(J(Q)), C\backslash(V \cup \{s_{c_h^*}\}))$$

$$\leq \Delta(P, C) - (\Delta(Z(J(Q)), C) - 9\Delta(Z(J(Q)), C^*) - \zeta(P, C, Q, V))$$

$$\leq (1 - \frac{1}{100k})\Delta(P, C),$$

where the first inequality follows from the fact that $s_{c_h^*} \notin V$ for each $P_h^* \in Q_S'$, the second inequality follows from $\Delta(P_h^*, \{s_{c_h^*}\}) \leq 9\Delta(P_h^*, \{c_h^*\})$ for each $P_h^* \in Q_S'$, and the last inequality follows from the definition of good $t$-clusters and $V \cup \{s_{c_h^*}\} = A'$. $\qquad\square$

**Lemma 7.** *Given a set $Q \in H_2$ of centers such that $J(Q)$ is a set of good $t$-clusters, the probability that step 4 of Algorithm 2 fails to sample a data point $q$ from $\chi(P_h^*)$ for each $P_h^* \in Q_S''$ in the $m(P_h^*)$-th iteration is at most $(1 + \frac{1}{30000k})e^{-\Delta(Z(Q_S''), C)/(300\Delta(P,C))}$.*

**Proof** We first consider an optimal cluster $P_h^* \in Q_S''$. Observe that $\Delta(P_h^*, C) < \frac{1}{30000k2^{t-1}}\Delta(P, C)$ by the definition of $Q_S''$. Define $p_f^h$ as the failure probability of not sampling data points from $\chi(P_h^*)$ in the $m(P_h^*)$-th iteration of step 4 in Algorithm 2. Then, we have that

$$p_f^h = 1 - \frac{\Delta(\chi(P_h^*), C)}{\Delta(P, C)} \geq 1 - \frac{\Delta(P_h^*, C)}{\Delta(P, C))} \geq 1 - \frac{1}{30000k2^{t-1}}.$$

Let $p_s^h = 1 - p_f^h$ be the success probability. Then, we have

$$p_s^h = 1 - p_f^h \leq (\frac{1}{1 - \frac{1}{30000k2^{t-1}}} - 1)p_f^h \leq \frac{\frac{1}{30000}}{k2^{t-1} - \frac{1}{30000}}p_f^h \leq \frac{1}{30000k(2^{t-1} - 1)}p_f^h,$$

where the last inequality follows from $\frac{1}{30000} < k$. Let $p_f(Q_S'')$ denote the probability of not sampling a data point from $\chi(P_h^*)$ for each $P_h^* \in Q_S''$ in the $m(P_h^*)$-th iteration of step 4 in Algorithm 2. For each $P_h^* \in Q_S''$, define $X_h = 1$ if the $m(P_h^*)$-th iteration of step 4 in Algorithm 2 samples a data point $q \in \chi(P_h^*)$, and $X_h = 0$ if the $m(P_h^*)$-th iteration fails to sample a data point $q \in \chi(P_h^*)$. Let $E_1$ be the event that $X_h = 0$ for each $P_h^* \in Q_S''$, and $E_2$ be the event that there are at least two clusters $P_h^*, P_j^* \in Q_S''$ such that $X_h \vee X_j = 1$ and $X_h \wedge X_j = 0$. Observe that there are at most $2^t - 2$ subcases if event $E_2$ happens. For each subcase $s$, let $\nu(s) = \{P_h^* : P_h^* \in Q_S'', X_h = 1\}$ and $\nu'(s) = \{P_h^* : P_h^* \in Q_S'', X_h = 0\}$, respectively. Define $P_r(s)$ as the probability that subcase $s$ happens. Since the sampling iterations in step 4 of Algorithm 2 are mutually independent, we have $P_r(s) = \prod_{P_h^* \in \nu(s)} p_s^h \prod_{P_h^* \in \nu'(s)} p_f^h$. By applying the inequality that $p_s^h \leq \frac{1}{30000k(2^{t-1}-1)}p_f^h$, we have

$$P_r(s) \leq (\frac{1}{30000k(2^{t-1} - 1)})^t \prod_{P_h^* \in Q_S''} p_f^h \leq \frac{1}{30000k(2^{t-1} - 1)} \prod_{P_h^* \in Q_S''} p_f^h$$

for each subcase $s$ of $E_2$, where the last inequality follows from $\frac{1}{30000k(2^{t-1}-1)} < 1$. Then, we have $P_r(E_2) \leq \frac{1}{30000k} \prod_{P_h^* \in Q_S''} p_f^h$ by taking a probability summation over all $2^t - 2$ subcases of event

$E_2$. Observe that the failure probability $p_f(Q_S'') = P_r(E_1) + P_r(E_2)$, where $P_r(E_1) = \prod_{P_h^* \in Q_S''} p_f^h$. Thus, we can get that

$$
\begin{aligned}
p_f(Q_S'') &\leq (1 + \frac{1}{30000k}) \prod_{P_h^* \in Q_S''} p_f^h = (1 + \frac{1}{30000k}) \prod_{P_h^* \in Q_S''} (1 - p_s^h) \\
&= (1 + \frac{1}{30000k}) \prod_{P_h^* \in Q_S''} (1 - \frac{\Delta(\chi(P_h^*), C)}{\Delta(P, C)}) \leq (1 + \frac{1}{30000k}) \prod_{P_h^* \in Q_S''} (1 - \frac{\Delta(P_h^*, C)}{300\Delta(P, C)}) \\
&\leq (1 + \frac{1}{30000k}) e^{-\Delta(Z(Q_S''), C)/300\Delta(P, C)},
\end{aligned}
$$

where the second to last inequality follows from Lemma 3, and the last inequality follows from the inequality $1 - x \leq e^{-x}$. □

**Lemma 8.** *For the optimal clusters in $H_G$, we have $\Delta(Z(H_G), C) \geq \frac{1}{100}\Delta(P, C)$.*

**Proof** Given any type-1 matched swap set $(Q, V)$, if $J(Q)$ is a set of good $t$-clusters with a type-1 matched swap set $(Q, V) \in M_2$, according to the assumption of subcase (2), we know that each optimal cluster $P_h^*$ in $Q_L$ is a bad single cluster with a swap pair $(c_h^*, c_l)$, where $c_l$ is a lonely center in $V$. Since $J(Q)$ is a set of good $t$-clusters, not all optimal clusters in $J(Q)$ are bad single clusters. Hence, for each optimal cluster $P_h^* \in Q_L$, we can find a lonely center $l(c_h^*) \in V$ such that $P_h^*$ is a bad single cluster with the swap pair $(c_h^*, l(c_h^*))$. By the definition of bad single cluster, we have

$$
\Delta(P_h^*, C) \leq \frac{1}{100k}\Delta(P, C) + 9\Delta(P_h^*, \{c_h^*\}) + \zeta(P, C, c_h^*, l(c_h^*)).
$$

For each optimal cluster $P_h^* \in Q_S'$, by the definition of $Q_S'$, it holds that $\Delta(P_h^*, C) \leq 3\Delta(P_h^*, \{c_h^*\})$. If $J(Q)$ is a set of bad $t$-clusters, by the definition of bad $t$-clusters, we also have

$$
\Delta(Z(J(Q)), C) \leq \frac{1}{100k}\Delta(P, C) + 9\Delta(Z(J(Q)), C^*) + \zeta(P, C, Q, V).
$$

Define $H_B^1 = \{c_h^* \in H_1 : P_h^* \text{ is a bad single cluster with the swap pair } (c_h^*, c_j) \in M_1\}$, $H_B^2 = \{Q \in H_2 : J(Q) \text{ is a set of bad } t\text{-clusters with the swap set } (Q, V) \in M_2\}$ and $H_{B'}^2 = \{Q_S' \cup Q_L : Q \in H_2, J(Q) \text{ is a set of good } t\text{-clusters with the swap set } (Q, V) \in M_2\}$, respectively. Putting all things together and taking a summation over all clusters in $H_B$, we can get that

$$
\begin{aligned}
\Delta(Z(H_B), C) &= \sum_{P_h^* \in H_B} \Delta(P_h^*, C) \\
&= \sum_{c_h^* \in H_B^1} \Delta(P_h^*, C) + \sum_{Q \in H_B^2} \Delta(Z(J(Q)), C) + \sum_{Q' \in H_{B'}^2} \Delta(Z(Q'), C) \\
&\leq \sum_{P_h^* \in H_B} \frac{1}{100k}\Delta(P, C) + 9\Delta(P_h^*, \{c_h^*\}) + 24\Delta(X(H_1'), C^*) + \frac{1}{5}\Delta(X(H_1'), C) \\
&\quad + 24\Delta(X(H_2'), C^*) + \frac{1}{5}\Delta(X(H_2'), C) \\
&\leq \frac{1}{100}\Delta(P, C) + 9\Delta(Z(H_B), C^*) + 24\Delta(X(H_1'), C^*) + \frac{1}{5}\Delta(X(H_1'), C) \\
&\quad + 24\Delta(X(H_2'), C^*) + \frac{1}{5}\Delta(X(H_2'), C),
\end{aligned}
$$

where the first inequality follows from Lemma 3. Then, together with Lemma 5, we can get that

$$\Delta(Z(H_G), C) \geq \Delta(P, C) - \Delta(Z(J(L)), C) - \Delta(Z(H_B), C)$$

$$\geq \Delta(P, C) - (1 + \frac{1}{t})(9\Delta(Z(J(L)), C^*) + 24\Delta(X(L'), C^*) + \frac{1}{5}\Delta(X(L'), C)$$

$$+ \frac{1}{100}\Delta(P, C) + \frac{1}{100}\Delta(P, C) + 9\Delta(Z(H_B), C^*) + 24\Delta(X(H_1'), C^*)$$

$$+ \frac{1}{5}\Delta(X(H_1'), C)) + 24\Delta(X(H_2'), C^*) + \frac{1}{5}\Delta(X(H_2'), C))$$

$$\geq \Delta(P, C) - (1 + \frac{1}{t})(\frac{1}{50}\Delta(P, C) + \frac{1}{5}\Delta(P, C) + 33Opt)$$

$$\geq \Delta(P, C) - \frac{33}{100}\Delta(P, C) - \frac{33}{50}\Delta(P, C)$$

$$\geq \frac{1}{100}\Delta(P, C),$$

where the second inequality follows from Lemma 5, the third inequality follows from the fact that $H_1' \cap H_2' \cap L' = \emptyset$ and $Z(H_B) \cap J(L) = \emptyset$, and the fourth inequality follows from $t \geq 2$ and the assumption that $\Delta(P, C) \geq 50(1 + \frac{1}{t})Opt$. □

**Theorem 2.** *In the $i$-th iteration of Algorithm 2, let $C'$ be the set of centers obtained in step 7. If the current clustering cost $\Delta(P, C)$ is larger than $50(1 + \frac{1}{t})Opt$, then with probability at least $\Omega(k^{-t})$, we have $\Delta(P, C') \leq (1 - \frac{1}{100k})\Delta(P, C)$. After $O(k^{O(t)} \log(\epsilon^{-1} \log k))$ iterations, we get an approximate solution with ratio $(50(1 + \frac{1}{t}) + \epsilon)$ in expectation.*

**Proof** Let $T = \delta k^{t+1} \log(24\epsilon^{-1} \log k)$, where $\delta$ is a sufficient large constant. Following the work in [11], we define another random process $X$ with initial clustering cost $\Delta(P, C')$ for a set $C'$ returned by the $k$-means++ algorithm such that for $T$ iterations of sampling and swaps, it reduces the value of $\Delta(P, C')$ by at least $(1 - \frac{1}{100k})$ with probability $\lambda k^{-t}$, and it increases the final value of $\Delta(P, C')$ by $50(1 + \frac{1}{t})OPT$, where $\lambda$ is a constant with $\lambda < \frac{\delta}{100}$. It is easy to see that $E[\Delta(P, C)] < E[X]$. Then, we have

$$E[X] = 50(1 + \frac{1}{t})Opt$$

$$+ \Delta(P, C') \sum_{i=1}^{T} \binom{T}{i} (\frac{1}{\lambda k^t})^i (1 - \frac{1}{\lambda k^t})^{T-i} (1 - \frac{1}{100k})^i$$

$$= \Delta(P, C')(1 - \frac{1}{100\lambda k^{t+1}})^T + 50(1 + \frac{1}{t})Opt$$

$$\leq \frac{\epsilon \Delta(P, C')}{24 \log k} + 50(1 + \frac{1}{t})Opt.$$

This implies that $E[\Delta(P, C)|C'] \leq \frac{\epsilon \Delta(P, C')}{24 \log k} + 50(1 + \frac{1}{t})Opt$. Then, we can get that

$$E[\Delta(P, C)] = \sum_{C'} E[\Delta(P, C)|C']P_r(C')$$

$$\leq \sum_{C'} P_r(C')(\frac{\epsilon \Delta(P, C')}{24 \log k} + 50(1 + \frac{1}{t})Opt)$$

$$\leq \frac{\epsilon E[\Delta(P, C')]}{24 \log k} + 50(1 + \frac{1}{t})Opt.$$

Since the $k$-means++ algorithm provides an approximation ratio of $8(\log k + 2)$ in expectation by Theorem 1, we have $E[\Delta(P, C)] \leq (50(1 + \frac{1}{t}) + \epsilon)Opt$. □

**Running Time Analysis.** By Theorem 2, in order to obtain a $(50(1 + \frac{1}{t}) + \epsilon)$-approximate solution, the iteration rounds for Algorithm 2 should be $O(k^{t+1} \log(\epsilon^{-1} \log k))$. In each iteration, it takes $O(ndk)$ time to update the distances between data points to their closest centers. During the sampling

process, $t$ data points are sampled according to the $D^2$-Sampling distribution to serve as the candidate set of centers for swapping in, which takes time $O(t)$ if the distances from data points to their centers are already known. It takes $O(k^t)$ time to enumerate each subset with size at most $t$ of the set of current centers opened. It takes $O(ndk^t)$ time to recalculate the clustering cost after each swap if $t$ nearest centers of each data point are maintained during the whole process. Thus, the total running time of Algorithm 2 is $O(ndk^{2t+1}\log(\epsilon^{-1}\log k))$.

## B  Complementary Experiments

### B.1  Experiments with Different Parameter Settings

In this section, we present the experiments on the performances of our proposed MLSP algorithm with different parameters $T$, $R'$, $\epsilon$ and $\eta$. We also compare our MLS and MLSP algorithms with LS++ algorithm using different iteration rounds to show the parameter robustness of different algorithms.

Tables 3, 4 and 5 show the comparison results of different algorithms with varying iteration rounds on dataset rds, KEGG and Urban_10, respectively. The results show that, our proposed MLSP algorithm always achieves the best clustering cost compared with LS++ and MLS algorithms. It can be seen that the number of iteration rounds has little impact on our MLSP algorithm. For MLS algorithm, it performs better on clustering cost compared with LS++ algorithm for most cases, which indicates that larger iteration rounds will not influence the performance of our algorithms.

Tables 6, 7, and 8 show the results of MLSP algorithm with varying number of failure upper bound $R'$ for fixed $\epsilon = 0.5$, $\eta = 0.5$ and $T = 400$ on dataset rds, KEGG and Urban_10, respectively. It can be seen that a larger failure upper bound will lead to smaller deviation, and the running time becomes higher. In general, the failure upper bound does not influence the performance of our proposed MLSP algorithm. Tables 9, 10, and 11 show the results of MLSP algorithm with varying parameters $\epsilon$ and $\eta$ for fixed $T = 400$ and $R' = 5$ on dataset rds, KEGG, and Urban_10, respectively. It can be seen that smaller values of $\epsilon$ and $\eta$ result in better performances on clustering costs with smaller deviation, and the running time becomes higher. There is no significant difference between different choices of parameters $\epsilon$ and $\eta$, which indicates that our proposed MLSP algorithm ensures the parameter robustness.

| Method | Rounds | Best Cost | Average Cost | Time(s) | Method | Rounds | Best Cost | Average Cost | Time(s) |
|---|---|---|---|---|---|---|---|---|---|
| LS++ | | 133.4074 | 139.6724±3.5172 | 2.56 | LS++ | | 133.1909 | 13582±2.0054 | 3.81 |
| MLS | 100 | 133.9806 | 139.6108±1.3052 | **1.12** | MLS | 200 | 132.5915 | 138.6551±2.3461 | **2.27** |
| MLSP | | **131.7410** | **132.9932±0.9546** | 29.02 | MLSP | | **131.7410** | **133.1258±2.2659** | 30.25 |
| LS++ | | 132.0117 | 137.2619±2.8543 | 5.49 | LS++ | | 132.1966 | 138.2526±2.0892 | 8.14 |
| MLS | 300 | 133.5915 | 138.6551±2.3461 | **3.55** | MLS | 400 | 131.8264 | 136.9465±3.1638 | **4.48** |
| MLSP | | **131.7410** | **132.3278±0.8694** | 42.67 | MLSP | | **131.7410** | **132.0323±0.4511** | 58.61 |
| LS++ | | 133.6918 | 140.7157±2.0026 | 12.26 | LS++ | | 131.7681 | 137.7633±3.0206 | 17.24 |
| MLS | 500 | 131.8298 | 138.3528±5.2943 | **5.57** | MLS | 600 | 131.8264 | 137.1135±3.6302 | **6.90** |
| MLSP | | **131.7410** | **132.8539±1.7382** | 66.73 | MLSP | | **131.7410** | **132.1285±0.5928** | 72.64 |
| LS++ | | 131.8519 | 137.8405±3.3180 | 22.83 | LS++ | | 131.8238 | 139.0674±2.5677 | 29.23 |
| MLS | 700 | 131.8493 | 136.1385±2.9510 | **7.66** | MLS | 800 | 131.7709 | 135.7753±4.0815 | **7.98** |
| MLSP | | **131.7410** | **132.1379±0.8516** | 81.88 | MLSP | | **131.7410** | **132.2103±0.7338** | 91.52 |

Table 3: Comparison results on dataset rds with varying iteration rounds

| Method | Rounds | Best Cost | Average Cost | Time(s) | Method | Rounds | Best Cost | Average Cost | Time(s) |
|---|---|---|---|---|---|---|---|---|---|
| LS++ | | 6.1871E+07 | 6.2712E+07±1.3E+06 | 2.13 | LS++ | | 6.1619E+07 | 6.3341E+07±1.8E+06 | 5.64 |
| MLS | 100 | 6.1587E+07 | 7.2593E+07±1.1E+07 | **2.11** | MLS | 200 | 6.1684E+07 | 6.4154E+07±2.2E+06 | **4.00** |
| MLSP | | **6.1534E+07** | **6.1559E+07±4.9E+04** | 22.22 | MLSP | | **6.1534E+07** | **6.1534E+07±0** | 36.08 |
| LS++ | | 6.1642E+07 | 6.2732E+07±1.7E+06 | 10.41 | LS++ | | 6.1547E+07 | 6.3276E+07±1.9E+06 | 16.58 |
| MLS | 300 | 6.1939E+07 | 6.5379E+07±7.2E+06 | **6.32** | MLS | 400 | 6.1547E+07 | 6.3255E+07±1.8E+06 | **7.98** |
| MLSP | | **6.1534E+07** | **6.1559E+07±4.9E+04** | 46.18 | MLSP | | **6.1534E+07** | **6.1546E+07±3.7E+04** | 57.13 |
| LS++ | | 6.1547E+07 | 6.2891E+07±1.6E+06 | 24.15 | LS++ | | 6.1547E+07 | 6.2878E+07±1.6E+06 | 32.97 |
| MLS | 500 | 6.1661E+07 | 6.3595E+07±1.7E+06 | **10.54** | MLS | 600 | **6.1534E+07** | 6.4042E+07±2.9E+06 | **12.95** |
| MLSP | | **6.1534E+07** | **6.1559E+07±4.9E+04** | 63.61 | MLSP | | **6.1534E+07** | **6.1572E+07±5.7E+04** | 77.49 |
| LS++ | | **6.1534E+07** | 6.2874E+07±1.3E+06 | 43.03 | LS++ | | 6.1547E+07 | 6.2261E+07±5.9E+05 | 54.28 |
| MLS | 700 | 6.1587E+07 | 6.5113E+07±3.4E+06 | **13.97** | MLS | 800 | 6.1953E+07 | 6.5383E+07+3.5E+06 | **16.22** |
| MLSP | | **6.1534E+07** | **6.1547E+07±3.7E+04** | 86.74 | MLSP | | **6.1534E+07** | **6.1559E+07±4.9E+04** | 96.73 |

Table 4: Comparison results on dataset KEGG with varying iteration rounds

| Method | Rounds | Best Cost | Average Cost | Time(s) | Method | Rounds | Best Cost | Average Cost | Time(s) |
|---|---|---|---|---|---|---|---|---|---|
| LS++ | | 24659.3733 | 25255.4895±613.5016 | 3.08 | LS++ | | 24660.9511 | 25024.9944±356.9319 | 8.36 |
| MLS | 100 | 24787.2318 | 25022.6100±348.1652 | **1.88** | MLS | 200 | 24659.3393 | 25022.5701±332.6797 | **3.34** |
| MLSP | | **24659.0733** | **24726.3142±134.6180** | 40.81 | MLSP | | **24659.0733** | **24806.2115±158.9739** | 48.55 |
| LS++ | | 24659.3733 | 25594.2717±914.0252 | 15.99 | LS++ | | **24659.0733** | 25383.4974±577.4748 | 25.79 |
| MLS | 300 | 24659.3733 | 25107.5051±420.6469 | **4.85** | MLS | 400 | 24659.3732 | 25166.5432±526.5936 | **6.31** |
| MLSP | | **24659.0733** | **24761.5254±176.6838** | 54.50 | MLSP | | **24659.0733** | **24793.3057±239.687** | 58.74 |
| LS++ | | 24659.3732 | 25217.6713±463.9049 | 37.94 | LS++ | | **24659.0733** | 25282.5551±727.8617 | 52.35 |
| MLS | 500 | 24659.3732 | 24835.8816±211.3455 | **7.79** | MLS | 600 | 24659.3732 | 24797.6361±28.3684 | **9.18** |
| MLSP | | **24659.0733** | **24762.4264±178.6638** | 73.87 | MLSP | | **24659.0733** | **24742.3513±134.4212** | 76.58 |
| LS++ | | 24659.3733 | 24852.2682±446.1149 | 69.19 | LS++ | | 24659.0844 | 25611.1218±891.9473 | 88.15 |
| MLS | 700 | 24659.4008 | 25073.7242±295.8309 | **10.61** | MLS | 800 | 24659.5079 | 25001.2324±351.8913 | **12.01** |
| MLSP | | **24659.0733** | **24784.8027±134.1987** | 81.40 | MLSP | | **24659.0733** | **24818.3757±219.2623** | 88.92 |

Table 5: Comparison results on dataset Urban_10 with varying iteration rounds

| Failure | Best Cost | Average Cost | Time(s) |
|---|---|---|---|
| 3 | 131.7408 | 133.8048±2.5306 | 39.49 |
| 4 | 131.7408 | 132.6853±1.5759 | 48.01 |
| 5 | 131.7408 | 132.8304±1.6403 | 59.41 |
| 6 | 131.7408 | 132.7752±1.1013 | 55.65 |
| 7 | 131.7408 | 132.2285±0.8758 | 60.03 |
| 8 | 131.7408 | 132.4954±0.6819 | 69.60 |
| 9 | 131.7408 | 132.3064±0.6889 | 72.11 |
| 10 | 131.7408 | 132.2246±0.6323 | 85.36 |

Table 6: Results of MLSP Algorithm on dataset rds with varying number of failure upper bound $R'$

| Failure | Best Cost | Average Cost | Time(s) |
|---|---|---|---|
| 3 | 6.1534E+07 | 6.1559E+07±4.9E+04 | 44.52 |
| 4 | 6.1534E+07 | 6.1559E+07±4.9E+04 | 59.61 |
| 5 | 6.1534E+07 | 6.1559E+07±4.9E+04 | 64.45 |
| 6 | 6.1534E+07 | 6.1559E+07±4.9E+04 | 70.69 |
| 7 | 6.1534E+07 | 6.1534E+07±0 | 76.37 |
| 8 | 6.1534E+07 | 6.1534E+07±0 | 91.05 |
| 9 | 6.1534E+07 | 6.1534E+07±0 | 100.82 |
| 10 | 6.1534E+07 | 6.1534E+07±0 | 105.89 |

Table 7: Results of MLSP Algorithm on dataset KEGG with varying number of failure upper bound $R'$

| Failure | Best Cost | Average Cost | Time(s) |
|---|---|---|---|
| 3 | 24659.0733 | 24824.3035±214.2367 | 41.44 |
| 4 | 24659.0733 | 24819.0528±238.5684 | 56.65 |
| 5 | 24659.0733 | 24755.9209±132.1597 | 59.97 |
| 6 | 24659.0733 | 24806.0814±158.7277 | 66.19 |
| 7 | 24659.0733 | 24829.4308±190.0205 | 80.24 |
| 8 | 24659.0733 | 24679.4947±58.5737 | 88.09 |
| 9 | 24659.0733 | 24703.7889±55.6132 | 92.57 |
| 10 | 24659.0733 | 24684.7586±51.1046 | 95.37 |

Table 8: Results of MLSP Algorithm on dataset Uran_10 with varying number of failure upper bound $R'$

| eta/epsilon | Best Cost | Average Cost | Time(s) |
|---|---|---|---|
| 0.25/0.25 | 131.7408 | 132.6835±1.9397 | 61.57 |
| 0.5/0.25 | 131.7408 | 133.5867±2.0429 | 58.97 |
| 0.75/0.25 | 131.7408 | 133.0508±1.7254 | 63.59 |
| 0.25/0.5 | 131.7408 | 132.3304±0.6313 | 50.81 |
| 0.5/0.5 | 131.7408 | 132.3244±1.5983 | 51.40 |
| 0.75/0.5 | 131.7408 | 132.6404±2.3554 | 50.50 |
| 0.25/0.75 | 131.7693 | 134.4831±2.2116 | 52.86 |
| 0.5/0.75 | 131.7408 | 132.3092±0.9371 | 61.99 |
| 0.75/0.75 | 131.7408 | 133.2093±2.1465 | 60.73 |

Table 9: Results of MLSP Algorithm on dataset rds with varying parameters $\epsilon$ and $\eta$

| eta/epsilon | Best Cost | Average Cost | Time(s) |
|---|---|---|---|
| 0.25/0.25 | 6.1534E+07 | 6.1572E+07±5.7E+04 | 85.99 |
| 0.5/0.25 | 6.1534E+07 | 6.1546E+07±3.7E+04 | 78.57 |
| 0.75/0.25 | 6.1534E+07 | 6.1572E+07±5.7E+04 | 72.15 |
| 0.25/0.5 | 6.1534E+07 | 6.1572E+07±5.7E+04 | 68.67 |
| 0.5/0.5 | 6.1534E+07 | 6.1546E+07±3.7E+04 | 65.05 |
| 0.75/0.5 | 6.1534E+07 | 6.1559E+07±4.9E+04 | 62.09 |
| 0.25/0.75 | 6.1534E+07 | 6.1559E+07±4.9E+04 | 65.15 |
| 0.5/0.75 | 6.1534E+07 | 6.1546E+07±3.7E+04 | 63.55 |
| 0.75/0.75 | 6.1534E+07 | 6.1546E+07±3.7E+04 | 64.35 |

Table 10: Results of MLSP algorithm on dataset KEGG with varying parameters $\epsilon$ and $\eta$

| eta/epsilon | Best Cost | Average Cost | Time(s) |
|---|---|---|---|
| 0.25/0.25 | 24659.0733 | 24681.3945±240.9711 | 66.64 |
| 0.5/0.25 | 24659.0733 | 24837.5858±203.3399 | 72.86 |
| 0.75/0.25 | 24659.0733 | 24786.8942±168.9089 | 59.95 |
| 0.25/0.5 | 24659.0733 | 24671.9425±38.3418 | 60.31 |
| 0.5/0.5 | 24659.0733 | 24756.0949±134.5952 | 63.27 |
| 0.75/0.5 | 24659.0733 | 24746.6868±146.2990 | 56.24 |
| 0.25/0.75 | 24659.0733 | 24719.8526±61.3117 | 57.11 |
| 0.5/0.75 | 24659.0733 | 24804.6754±173.0019 | 54.95 |
| 0.75/0.75 | 24659.0733 | 24753.6144±204.2255 | 56.56 |

Table 11: Results of MLSP algorithm on dataset Urban_10 with varying parameters $\epsilon$ and $\eta$

## B.2 Experiments on Small Datasets

In this section, we present the experiments of the performances of our proposed MLSP algorithm on other datasets with sizes smaller than $50,000$ used in [15].

Table 12 and Table 13 show the results on 18 datasets with sizes smaller than $50,000$ using $k = 10$. It can be seen that, on each dataset, the best clustering cost returned by our MLSP algorithm is smaller than BB method (note that BB method already guarantees a gap smaller than 0.1% to the optimal solutions on small datasets) and other local search methods. On each dataset, the average clustering cost returned by our MLSP algorithm nearly matches the result of the BB method. As for FLS, although the performance of FLS is better than that of LS++, our proposed MLSP algorithm improves the performance of FLS on clustering cost with smaller deviation for most datasets. As for running time, there is no significant difference on the running time among different local search algorithms for small datasets.

| Method | Size | BB(Cost) | Best | Mean | Time(s) |
|---|---|---|---|---|---|
| LS++ |  |  | 30.01 | 30.45±0.53 | **0.04** |
| FLS |  |  | 29.79 | 29.91±0.14 | 0.15 |
| LLOYD | Iris(150*4) | 29.79(735s) | 30.32 | 30.90±0.4 | **0.01** |
| MLSP |  |  | **29.74** | **29.76±0.33** | 6.61 |
| MLS |  |  | 29.93 | 30.21±0.19 | 0.17 |
| LS++ |  |  | 216.36 | 219.43±2.84 | **0.37** |
| FLS |  |  | **214.52** | 216.68±2.83 | 1.78 |
| LLOYD | SEEDS(210*7) | 218.49(448s) | 215.65 | 219.99±2.93 | **0.12** |
| MLSP |  |  | **214.52** | **215.08±0.49** | 8.07 |
| MLS |  |  | 214.95 | 219.29±2.82 | 0.46 |
| LS++ |  |  | **251.86** | 254.17±1.71 | **0.37** |
| FLS |  |  | **251.86** | 252.05±0.56 | 1.61 |
| LLOYD | GLASS(214*9) | 251.86(2566s) | 253.25 | 259.94±3.94 | **0.11** |
| MLSP |  |  | **251.86** | **251.99±0.54** | 7.25 |
| MLS |  |  | 253.29 | 254.00±0.55 | 0.46 |
| LS++ |  |  | 377753 | 392980±4437 | **0.37** |
| FLS |  |  | **375974** | 377496±1934 | 1.71 |
| LLOYD | BM(249*6) | 375974(1204s) | 376590 | 384475±5560 | **0.07** |
| MLSP |  |  | **375974** | **376276±265** | 6.81 |
| MLS |  |  | 378649 | 384982±4131 | 0.37 |
| LS++ |  |  | 29.57 | 30.16±0.29 | **0.38** |
| FLS |  |  | **29.27** | 29.38±0.15 | 1.73 |
| LLOYD | UK(258*5) | 29.28(4h) | 29.94 | 30.33±0.26 | **0.08** |
| MLSP |  |  | **29.27** | **29.29±0.01** | 7.14 |
| MLS |  |  | 29.41 | 30.00±0.27 | 0.39 |
| LS++ |  |  | **6.96E+10** | 7.01E+10±5.7E+08 | **0.39** |
| FLS |  |  | **6.96E+10** | 7.06E+10±3.1E+09 | 1.77 |
| LLOYD | HF(299*12) | 6.96E+10(4h) | **6.96E+10** | 6.97E+10±2.3E+08 | **0.09** |
| MLSP |  |  | **6.96E+10** | **6.96E+10±0** | 8.56 |
| MLS |  |  | **6.96E+10** | 7.01E+10±4.3E+08 | 0.4 |
| LS++ |  |  | **3.36E+10** | 3.45E+10±6.0E+08 | **0.4** |
| FLS |  |  | **3.36E+10** | 3.41E+10±4.5E+08 | 2.01 |
| LLOYD | WHO(440*8) | 3.40E+10(4h) | 3.44E+10 | 3.49E+10±7.4E+08 | **0.11** |
| MLSP |  |  | **3.36E+10** | **3.37E+10±4.6E+07** | 9.17 |
| MLS |  |  | 3.42E+10 | 3.52E+10±5.6E+08 | **0.4** |
| LS++ |  |  | 1.1315E+06 | 1.1454E+06±8.7E+03 | **0.45** |
| FLS |  |  | 1.1312E+06 | 1.1412E+06±2.4E+04 | 1.81 |
| LLOYD | HCV(572*12) | 1.1315E+06(4h) | 1.1329E+06 | 1.1538E+06±2.9E+04 | **0.15** |
| MLSP |  |  | **1.1311E+06** | **1.1410E+06±2.2E+04** | 11.05 |
| MLS |  |  | 1.1505E+06 | 1.2135E+06±4.2E+04 | 0.9 |
| LS++ |  |  | **1.0786E+06** | 1.0983E+06±1.1E+04 | 0.51 |
| FLS |  |  | **1.0786E+06** | 1.0816E+06±5.3E+03 | 2.11 |
| LLOYD | Abs(740*21) | 1.0786E+06(4h) | 1.0824E+06 | 1.1209E+06±2.1E+04 | **0.12** |
| MLSP |  |  | **1.0786E+06** | **1.0786E+06±0** | 13.42 |
| MLS |  |  | **1.0789E+06** | 1.0979E+05±1.1E+04 | **0.47** |

Table 12: Comparison results on clustering costs and running time with $k = 10$ on datasets with sizes ranging from 150 to 740

| Method | Size | BB(Cost) | Best | Mean | Time(s) |
|---|---|---|---|---|---|
| LS++ |  |  | 776.04 | 793.74±7.3 | **0.51** |
| FLS |  |  | **762.16** | 766.38±4.9 | 2.14 |
| LLOYD | TR(980*10) | 772.47(4h) | 770.72 | 776.48±5.0 | **0.12** |
| MLSP |  |  | **762.16** | **764.71±1.9** | 11.89 |
| MLS |  |  | 785.01 | 805.89±11.5 | 0.56 |
| LS++ |  |  | **1.1734E+08** | 1.2124E+08±3.4E+05 | **0.58** |
| FLS |  |  | **1.1734E+08** | 1.2092E+08±3.9E+05 | 2.35 |
| LLOYD | SGC(1000*21) | 1.1742E+08(4h) | **1.1734E+08** | **1.1749E+08±1.7E+05** | **0.09** |
| MLSP |  |  | **1.1734E+08** | 1.1752E+08±5.1E+04 | 13.45 |
| MLS |  |  | 1.1735E+08 | 1.1846E+08±9.6E+04 | 0.62 |
| LS++ |  |  | 2.7116E+06 | 2.7657E+06±4.4E+04 | **0.58** |
| FLS |  |  | 2.7073E+06 | 2.8519E+06±1.2E+04 | 2.51 |
| LLOYD | HEMI(1995*7) | 2.7421E+06(4h) | 2.7144E+06 | 2.7349E+06±6.7E+03 | **0.18** |
| MLSP |  |  | **2.7070E+06** | **2.7123E+06±6.7E+03** | 13.39 |
| MLS |  |  | 2.7292E+06 | 2.7886E+06±6.9E+04 | 0.59 |
| LS++ |  |  | 5.3689E+09 | 5.4772E+09±1.0E+08 | **0.58** |
| FLS |  |  | 5.3610E+09 | 5.4256E+09±5.6E+07 | 2.65 |
| LLOYD | pr2392(2392*2) | 5.3578E+09(4h) | 5.3599E+09 | **5.3624E+09±2.3E+07** | **0.14** |
| MLSP |  |  | **5.3578E+09** | 5.3668E+09±2.3E+07 | 10.76 |
| MLS |  |  | 5.3629E+09 | 5.4203E+09±6.2E+07 | **0.58** |
| LS++ |  |  | 1.3902E+05 | 1.4155E+05±1.5E+03 | **0.62** |
| FLS |  |  | 1.3829E+05 | 1.4018E+05±1.1E+03 | 3.43 |
| LLOYD | TRR(5456*24) | 1.3796E+05(4h) | 1.3868E+05 | 1.4177E+05±2.1E+03 | **0.18** |
| MLSP |  |  | **1.3796E+05** | **1.3829E+05±2.5E+02** | 25.48 |
| MLS |  |  | 1.4591E+05 | 1.4939E+05±2.0E+03 | 0.64 |
| LS++ |  |  | 1167.6 | 1184.01±15.88 | **0.58** |
| FLS |  |  | **1163.7** | 1174.1±13.06 | 5.06 |
| LLOYD | AC(7195*22) | 1181.7(4h) | 1169 | 1172.1±6.05 | **0.36** |
| MLSP |  |  | **1163.7** | **1168.2±9.01** | 22.32 |
| MLS |  |  | 1165.5 | 1181.7±12.8 | 0.64 |
| LS++ |  |  | 1.6104E+06 | 1.6508E+06±2.4E+05 | **0.24** |
| FLS |  |  | **1.6099E+06** | 1.6196E+06±7.3E+04 | 2.26 |
| LLOYD | rds_cnt(10000*4) | 1.6119E+06(4h) | **1.6009E+06** | 1.6247E+06±1.4E+05 | **0.11** |
| MLSP |  |  | **1.6009E+06** | **1.6105E+06±7.2E+02** | 18.67 |
| MLS |  |  | 1.6146E+06 | 1.6520E+06±2.5E+05 | 0.42 |
| LS++ |  |  | 1.8286E+07 | 1.8421E+07±1.1E+04 | **0.28** |
| FLS |  |  | 1.8269E+07 | 1.8404E+07±1.3E+04 | 0.94 |
| LLOYD | HTRU2(17898*8) | 1.8723E+07(4h) | **1.8266E+07** | 1.8380E+07±2.0E+05 | **0.15** |
| MLSP |  |  | **1.8266E+07** | **1.8312E+07±4.2E+03** | 24.43 |
| MLS |  |  | 1.8275E+07 | 1.8569E+07±1.7E+04 | 0.3 |
| LS++ |  |  | 9.0239E+06 | 9.1784E+06±9.3E+04 | 7.06 |
| FLS |  |  | **8.9904E+06** | 9.0384E+06±5.4E+04 | 19.87 |
| LLOYD | GT(36733*11) | 8.9909E+06(4h) | 9.0001E+06 | 9.1229E+06±7.7E+04 | **0.33** |
| MLSP |  |  | **8.9904E+06** | **9.0131E+06±3.6E+04** | 47.51 |
| MLS |  |  | 9.0239E+06 | 9.1784e+06±9.3E+04 | **6.29** |

Table 13: Comparison results on clustering costs and running time with $k = 10$ on datasets with sizes ranging from 980 to 36733

## B.3 Experiments with Different Values of $k$

In this section, we present the experiments of the performances of our proposed MLSP and MLS algorithms on different datasets with different values of $k$. Tables 14 and 15 present the performances of different local search algorithms with $k = 3$ on small datasets. Table 18 presents the performances of different local search algorithms with $k = 3$ on large datasets. Tables 16 and 17 present the performances of different local search algorithms with $k = 5$ on small datasets. Table 19 presents the performances of different local search algorithms with $k = 5$ on large datasets.

It can be seen that, compared with other local search methods, our proposed MLSP algorithm achieves the best clustering performance on most datasets with different values of $k$. On each dataset, the best clustering cost returned by our MLSP algorithm matches the clustering cost of BB method (note that BB method already guarantees a gap of $0.1\%$ to the optimal solution). On each dataset, the average clustering cost returned by our MLSP algorithm nearly matches the result of the BB method. As for running time, there is no significant difference on running time among all local search algorithms on small datasets. However, as the data sizes grow, our proposed MLS algorithm becomes much faster than other local search algorithms.

| Method | Size | BB(Cost) | Best | Mean | Time(s) |
|---|---|---|---|---|---|
| LS++ | | | **83.96** | 84.82±1.05 | **0.26** |
| FLS | | | **83.96** | **83.96±0** | 0.5 |
| LLOYD | Iris(150*4) | 83.96(93s) | 84.68 | 84.68±0 | **0.09** |
| MLSP | | | **83.96** | **83.96±0** | 3.46 |
| MLS | | | **83.96** | 84.52±0.73 | 0.43 |
| LS++ | | | **598.29** | 600.83±5.59 | **0.27** |
| FLS | | | **598.29** | **598.29±0** | 0.51 |
| LLOYD | SEEDS(210*7) | 598.29(84s) | **598.29** | 598.30±0.69 | **0..07** |
| MLSP | | | **598.29** | **598.29±0** | 3.34 |
| MLS | | | **598.29** | 600.70±5.63 | 0.67 |
| LS++ | | | **629.02** | 629.51±0.73 | **0.27** |
| FLS | | | **629.02** | **629.02±0** | 0.62 |
| LLOYD | GLASS(214*9) | 692.02(107s) | **629.02** | **629.02±0** | **0.05** |
| MLSP | | | **629.02** | **629.02±0** | 3.37 |
| MLS | | | **629.02** | 630.72±5.08 | 0.68 |
| LS++ | | | **8.63E+05** | 8.69E+05±6.4E+03 | **0.27** |
| FLS | | | **8.63E+05** | **8.63E+05±0** | 0.57 |
| LLOYD | BM(249*6) | 8.63E+05(86s) | 8.65E+05 | 8.65E+05±0 | **0.03** |
| MLSP | | | **8.63E+05** | **8.63E+05±0** | 3.69 |
| MLS | | | **8.63E+05** | 8.74E+05±7.3E+03 | 0.67 |
| LS++ | | | 50.92 | 51.95±0.71 | **0.29** |
| FLS | | | **50.77** | 50.83±0.12 | 0.54 |
| LLOYD | UK(258*5) | 50.77(89s) | 51.29 | 51.31±0.03 | **0.04** |
| MLSP | | | **50.77** | **50.77±0** | 3.67 |
| MLS | | | **50.77** | 51.78±0.81 | 0.67 |
| LS++ | | | **7.83E+11** | 7.90E+11±6.5E+10 | **0.27** |
| FLS | | | **7.83E+11** | 7.84E+11±2.0E+10 | 0.56 |
| LLOYD | HF(299*12) | 7.83E+11(107s) | **7.83E+11** | 7.83E+11±6.3E+07 | **0.05** |
| MLSP | | | **7.83E+11** | **7.83E+11±3.3E+07** | 4.02 |
| MLS | | | **7.83E+11** | 7.86E+11±3.4E+09 | 0.74 |
| LS++ | | | **8.33E+10** | 8.43E+10±6.7E+08 | **0.29** |
| FLS | | | **8.33E+10** | **8.33E+10±0** | 0.58 |
| LLOYD | WHO(440*8) | 8.33E+10(117s) | 8.40E+10 | 8.40E+10±0 | **0.06** |
| MLSP | | | **8.33E+10** | **8.33E+10±0** | 3.74 |
| MLS | | | **8.33E+10** | 8.46E+10±8.1E+08 | 0.71 |
| LS++ | | | **2.75E+06** | **2.75E+06±0** | **0.32** |
| FLS | | | **2.75E+06** | 2.85E+06±1.8E+05 | 0.61 |
| LLOYD | HCV(572*12) | 2.75E+06(215s) | **2.75E+06** | 2.79E+06±4.7E+04 | **0.08** |
| MLSP | | | **2.75E+06** | **2.75E+06±0** | 4.04 |
| MLS | | | **2.75E+06** | 2.79E+06±6.8E+04 | 0.78 |
| LS++ | | | 2.63E+06 | 2.70E+06±5.8E+04 | **0.33** |
| FLS | | | **2.62E+06** | **2.62E+06±0** | 0.69 |
| LLOYD | Abs(740*21) | 2.62E+06(119s) | **2.62E+06** | **2.62E+06±0** | **0.07** |
| MLSP | | | **2.62E+06** | **2.62E+06±0** | 5.4 |
| MLS | | | **2.62E+06** | 2.66E+06±2.1E+04 | 0.81 |

Table 14: Comparison results on clustering costs and running time with $k = 3$ on datasets with sizes ranging from 150 to 740

| Method | Size | BB(Cost) | Best | Mean | Time(s) |
|---|---|---|---|---|---|
| LS++ | | | 1145.77 | 1159.73±10.48 | **0.33** |
| FLS | | | **1134.45** | **1134.45±0** | 0.56 |
| LLOYD | TR(980*10) | 1134.45(126s) | 1136.93 | 1138.09±0.94 | **0.09** |
| MLSP | | | **1134.45** | **1134.45±0** | 4.75 |
| MLS | | | **1134.45** | 1154.49±14.61 | 0.78 |
| LS++ | | | **1.28E+09** | 1.28E+09±1.79E+06 | **0.35** |
| FLS | | | **1.28E+09** | **1.28E+09±0** | 0.82 |
| LLOYD | SGC(1000*21) | 1.28E+09(140s) | **1.28E+09** | **1.28E+09±0** | **0.13** |
| MLSP | | | **1.28E+09** | **1.28E+09±0** | 5.18 |
| MLS | | | **1.28E+09** | 1.28E+09±1.1E+06 | 0.81 |
| LS++ | | | **9.91E+06** | 9.91E+06±2.5E+03 | **0.53** |
| FLS | | | **9.91E+06** | **9.91E+06±0** | 0.92 |
| LLOYD | HEMI(1995*7) | 9.91E+06(97s) | **9.91E+06** | **9.91E+06±0** | **0.09** |
| MLSP | | | **9.91E+06** | **9.91E+06±0** | 5.44 |
| MLS | | | **9.91E+06** | 9.91E+06±2.3E+03 | 0.79 |
| LS++ | | | **2.13E+09** | 2.15E+09±1.7E+08 | **0.48** |
| FLS | | | **2.13E+09** | 2.14E+09±1.4E+08 | 0.89 |
| LLOYD | pr2392(2392*2) | 2.13E+09(123s) | **2.13E+09** | **2.13E+09±0** | **0.09** |
| MLSP | | | **2.13E+09** | **2.13E+09±0** | 5.92 |
| MLS | | | **2.13E+09** | 2.15E+09±1.5E+08 | 0.76 |
| LS++ | | | **1.96E+05** | 1.98E+05±1.7E+03 | **0.49** |
| FLS | | | **1.96E+05** | 1.97E+05±6.8E+02 | 0.97 |
| LLOYD | TRR(5456*24) | 1.96E+05(325s) | **1.96E+05** | 2.00E+05±3.3E+03 | **0.09** |
| MLSP | | | **1.96E+05** | **1.96E+05±3.8E+02** | 6.06 |
| MLS | | | **1.96E+05** | 1.98E+05±1.7E+03 | 0.78 |
| LS++ | | | **2199.1** | 2234.68±41.91 | **0.92** |
| FLS | | | **2199.1** | **2201.90±8.42** | 2.02 |
| LLOYD | AC(7195*22) | 2199.10(222s) | 2227.18 | 2227.18±0 | **0.16** |
| MLSP | | | **2199.1** | 2205.75±19.95 | 6.43 |
| MLS | | | **2199.1** | 2212.69±26.49 | 1.06 |
| LS++ | | | **1.49E+07** | 1.49E+07±269.4 | 1.1 |
| FLS | | | **1.49E+07** | **1.49E+07±0** | 1.91 |
| LLOYD | rds_cnt(10000*4) | 1.49E+07(203s) | **1.49E+07** | 1.49E+07±331.2 | **0.03** |
| MLSP | | | **1.49E+07** | **1.49E+07±0** | 4.96 |
| MLS | | | **1.49E+07** | 1.49E+07±226.1 | **0.91** |
| LS++ | | | **8.21E+07** | 8.22E+07±1.6E+05 | **1.65** |
| FLS | | | **8.21E+07** | **8.21E+07±0** | 3.71 |
| LLOYD | HTRU2(17898*8) | 8.21E+07(1555s) | **8.21E+07** | 8.21E+07±7.9E+03 | **0.97** |
| MLSP | | | **8.21E+07** | **8.21E+07±0** | 8.44 |
| MLS | | | **8.21E+07** | 8.22E+07±2.0E+05 | 1.21 |
| LS++ | | | **1.95E+07** | 1.97E+07±4.9E+05 | 4.38 |
| FLS | | | **1.95E+07** | 1.95E+07±6.0E+04 | 7.28 |
| LLOYD | GT(36733*11) | 1.95E+07(1936s) | **1.95E+07** | 1.99E+07±5.4E+05 | **1.18** |
| MLSP | | | **1.95E+07** | **1.95E+07±7.3E+03** | 12.83 |
| MLS | | | **1.95E+07** | 1.97E+07±4.7E+05 | **3.82** |

Table 15: Comparison results on clustering costs and running time with $k = 3$ on datasets with sizes ranging from 980 to 36733

| Method | Size | BB(Cost) | Best | Mean | Time(s) |
|---|---|---|---|---|---|
| LS++ | | | **50.97** | 52.75±1.74 | **0.37** |
| FLS | | | **50.97** | 51.70±1.10 | 0.78 |
| LLOYD | Iris(150*4) | 50.92(355s) | 51.19 | 51.19±0 | **0.1** |
| MLSP | | | **50.97** | **50.98±0.02** | 3.53 |
| MLS | | | **50.97** | 51.78±1.11 | 0.81 |
| LS++ | | | **401.21** | 404.58±3.30 | **0.3** |
| FLS | | | **401.21** | 402.46±1.25 | 0.81 |
| LLOYD | SEEDS(210*7) | 401.21(376s) | **401.21** | 406.69±2.23 | **0.11** |
| MLSP | | | **401.21** | **401.46±0.75** | 3.62 |
| MLS | | | **401.21** | 409.65±9.40 | 0.83 |
| LS++ | | | 437.88 | 443.63±7.54 | **0.29** |
| FLS | | | **437.73** | 439.82±6.29 | 0.89 |
| LLOYD | GLASS(214*9) | 437.73(592s) | 437.88 | 457.22±15.78 | **0.07** |
| MLSP | | | **437.73** | **437.73±0** | 3.43 |
| MLS | | | **437.73** | 439.37±2.49 | 0.86 |
| LS++ | | | **6.0249E+05** | 6.0213E+05±1.8E+03 | **0.29** |
| FLS | | | **6.0249E+05** | 6.1293E+05±5.2E+03 | 0.83 |
| LLOYD | BM(249*6) | 6.0249E+05(389s) | **6.0249E+05** | 6.1293E+05±4.9E+03 | **0.04** |
| MLSP | | | **6.0249E+05** | **6.0249E+05±0** | 3.71 |
| MLS | | | **6.0249E+05** | 6.2224E+05±2.0E+04 | 0.83 |
| LS++ | | | **40.17** | 40.97±0.63 | **0.29** |
| FLS | | | **40.17** | 40.28±0.25 | 0.82 |
| LLOYD | UK(258*5) | 40.17(457s) | 40.24 | 41.16±0.57 | **0.06** |
| MLSP | | | **40.17** | **40.25±0.25** | 3.72 |
| MLS | | | 40.34 | 41.55±0.85 | 0.83 |
| LS++ | | | **3.0998E+11** | 3.1257E+11±2.5E+10 | **0.31** |
| FLS | | | **3.0998E+11** | 3.1083E+11±1.1E+10 | 0.89 |
| LLOYD | HF(299*12) | 3.0998E+11(1723s) | **3.0998E+11** | **3.1001E+11±1.2E+08** | **0.06** |
| MLSP | | | **3.0998E+11** | 3.1021E+11±7.1E+08 | 4.65 |
| MLS | | | **3.0998E+11** | 3.1499E+11±5.6E+09 | 0.92 |
| LS++ | | | 5.5982E+10 | 5.6642E+10±3.6E+08 | **0.33** |
| FLS | | | **5.5914E+10** | 5.6333E+10±2.1E+08 | 0.89 |
| LLOYD | WHO(440*8) | 5.5914E+10(1840s) | **5.5914E+10** | 5.6219E+10±1.25E+08 | **0.09** |
| MLSP | | | **5.5914E+10** | **5.6034E+10±1.25E+08** | 4.84 |
| MLS | | | 5.5973E+10 | 5.6460E+10±3.5E+08 | 0.88 |
| LS++ | | | **1.9716E+06** | 1.9722E+06±622.78 | **0.34** |
| FLS | | | **1.9716E+06** | 2.0044E+06±2.6E+04 | 0.91 |
| LLOYD | HCV(572*12) | 1.9716E+06(12768s) | **1.9716E+06** | 1.9995E+06±3.3E+04 | **0.13** |
| MLSP | | | **1.9716E+06** | **1.9716E+06±0** | 5.02 |
| MLS | | | **1.9716E+06** | 2.0113E+06±3.1E+04 | 0.93 |
| LS++ | | | **1.7472E+06** | 1.7630E+06±3.8E+04 | **0.36** |
| FLS | | | **1.7472E+06** | **1.7472E+06±0** | 0.98 |
| LLOYD | Abs(740*21) | 1.7472E+06(410s) | **1.7472E+06** | 1.7502E+06±1.5E+03 | **0.09** |
| MLSP | | | **1.7472E+06** | **1.7472E+06±0** | 6.03 |
| MLS | | | **1.7472E+06** | 1.7676E+06±3.5E+04 | 1.16 |

Table 16: Comparison results on clustering costs and running time with $k = 5$ on datasets with sizes ranging from 150 to 740

| Method | Size | BB(Cost) | Best | Mean | Time(s) |
|---|---|---|---|---|---|
| LS++ | | | 966.64 | 990.43±14.69 | **0.35** |
| FLS | | | **953.39** | 956.16±3.43 | 1.13 |
| LLOYD | TR(980*10) | 953.39(824s) | 965.74 | 969.10+5.65 | **0.14** |
| MLSP | | | **953.39** | **954.62±2.48** | 6.28 |
| MLS | | | 983.22 | 1000.86±10.34 | 0.93 |
| LS++ | | | 4.6926E+08 | 4.7268E+08±2.8E+06 | **0.39** |
| FLS | | | **4.6919E+08** | 4.7197E+08±2.1E+06 | 1.06 |
| LLOYD | SGC(1000*21) | 4.6919E+08(3290s) | **4.6919E+08** | **4.6949E+08±2.3E+05** | **0.09** |
| MLSP | | | **4.6919E+08** | 4.6973E+08±1.0E+06 | 7.7 |
| MLS | | | 4.7204E+08 | 4.7529E+08±2.7E+06 | 1.22 |
| LS++ | | | **5.3811E+06** | 5.4497E+06±5.6E+04 | **0.45** |
| FLS | | | **5.3811E+06** | 5.4090E+06±3.4E+04 | 1.24 |
| LLOYD | HEMI(1995*7) | 5.3811E+06(930s) | **5.3811E+06** | 5.4262E+06±1.5E+04 | **0.13** |
| MLSP | | | **5.3811E+06** | **5.3880E+06±2.1E+04** | 7.04 |
| MLS | | | **5.3811E+06** | 5.4781E+06±1.4E+04 | 0.86 |
| LS++ | | | 1.1621E+09 | 1.1700E+09±6.3E+07 | **0.44** |
| FLS | | | **1.1619E+09** | 1.1636E+09±2.3E+07 | 1.31 |
| LLOYD | pr2392(2392*2) | 1.1619E+09(625s) | 1.1620E+09 | 1.1620E+09±2.9E+05 | **0.14** |
| MLSP | | | **1.1619E+09** | **1.1619E+09±0** | 7.46 |
| MLS | | | 1.1627E+09 | 1.1759E+09±1.4E+07 | 0.81 |
| LS++ | | | 1.6920E+05 | 1.7055E+05±626.4 | **0.54** |
| FLS | | | 1.6894E+05 | 1.7012E+05±529.7 | 1.92 |
| LLOYD | TRR(5456*24) | 1.6870E+05(3094s) | 1.7086E+05 | 1.7250E+05±3.0E+03 | **0.15** |
| MLSP | | | **1.6870E+05** | **1.6903E+05±473.9** | 9.57 |
| MLS | | | 1.6914E+05 | 1.7027E+05±548.1 | 0.89 |
| LS++ | | | **1636.1** | 1636.93±0.99 | **0.67** |
| FLS | | | **1636.1** | **1636.14±0** | 2.6 |
| LLOYD | AC(7195*22) | 1636.1(3552s) | 1542.6 | 1642.56±0 | **0.24** |
| MLSP | | | **1636.1** | **1636.14±0** | 7.68 |
| MLS | | | **1636.1** | 1637.25±1.26 | 1.57 |
| LS++ | | | 5.3728E+06 | 5.3747E+06±2.2E+03 | **1.29** |
| FLS | | | **5.3725E+06** | **5.3725E+06±103** | 3.11 |
| LLOYD | rds_cnt(10000*4) | 5.3725E+06(7171s) | 5.3744E+06 | 5.3751E+06±987 | **0.05** |
| MLSP | | | **5.3725E+06** | 5.3728E+06±554 | 7.53 |
| MLS | | | 5.3728E+06 | 5.3742E+06±2.1E+03 | 1.58 |
| LS++ | | | 4.2176E+07 | 4.2260E+07±4.6E+04 | 2.31 |
| FLS | | | **4.2152E+07** | 4.2160E+07±9.3E+03 | 5.03 |
| LLOYD | HTRU2(17898*8) | 4.2154E+07(4h) | 4.2171E+07 | 4.2173E+07±2.9E+03 | **0.16** |
| MLSP | | | **4.2152E+07** | **4.2153E+07±2.9E+03** | 11.01 |
| MLS | | | 4.2268E+07 | 4.2413E+07±1.4E+04 | **1.56** |
| LS++ | | | **1.3351E+07** | 1.3643E+07±3.8E+05 | 5.34 |
| FLS | | | **1.3351E+07** | 1.3438E+07±2.6E+05 | 10.67 |
| LLOYD | GT(36733*11) | 1.3351E+07(4h) | **1.3351E+07** | 1.3530E+07±3.4E+05 | **0.19** |
| MLSP | | | **1.3351E+07** | **1.3351E+07±0** | 16.61 |
| MLS | | | 1.3358E+07 | 1.3673E+07±3.9E+05 | **4.93** |

Table 17: Comparison results on clustering costs and running time with $k = 5$ on datasets with sizes ranging from 980 to 36733

| Method | Size | BB(Cost) | Best | Mean | Time(s) |
|---|---|---|---|---|---|
| LS++ | | | 476.88 | 488.32±22.61 | 2.93 |
| FLS | | | 476.88 | 476.88±0 | 10.44 |
| LLOYD | rds(50,000*3) | 476.79(811s) | 476.88 | 507.69±31.3 | **0.18** |
| MLSP | | | **476.79** | **476.8±0.07** | 15.52 |
| MLS | | | 476.88 | 482.77±16.93 | **2.26** |
| LS++ | | | **4.9412E+08** | **4.9412E+08±0** | **1.04** |
| FLS | | | **4.9412E+08** | **4.9412E+08±0** | 7.27 |
| LLOYD | KEGG(53,413*23) | 4.9412E+08(3901s) | **4.9412E+08** | **4.9412E+08±0** | 0.26 |
| MLSP | | | **4.9412E+08** | **4.9412E+08±0** | 12.89 |
| MLS | | | **4.9412E+08** | **4.9412E+08±0** | 1.97 |
| LS++ | | | **1.15E+05** | 1.16E+05±1.7E+03 | 3.26 |
| FLS | | | **1.15E+05** | 1.15E+05±229 | 14.73 |
| LLOYD | Urban_10(100,000*2) | 1.15E+05(6834s) | **1.15E+05** | 1.23E+05±1.1E+04 | **0.18** |
| MLSP | | | **1.15E+05** | **1.15E+05±72** | 26.14 |
| MLS | | | **1.15E+05** | 1.16E+05±1.6E+03 | **3.08** |
| LS++ | | | **1.6396E+15** | 1.6398E+15±3.3E+11 | 8.27 |
| FLS | | | **1.6396E+15** | 1.6397E+15±1.3E+11 | 58.78 |
| LLOYD | RNG_AGR(199,843*7) | 1.64E+15(4h) | **1.6396E+15** | 1.6398E+15±3.2E+11 | **0.42** |
| MLSP | | | **1.6396E+15** | **1.6396E+15±0** | 53.25 |
| MLS | | | **1.6396E+15** | 1.6398E+15±1.3E+11 | **7.73** |
| LS++ | | | **414253** | 417302±4629 | 9.86 |
| FLS | | | **414253** | 414733±956 | 50.11 |
| LLOYD | Urban_GB(360,177*2) | OOM | **414253** | 423509±12613 | **0.32** |
| MLSP | | | **414253** | **414642±314** | 71.46 |
| MLS | | | **414253** | 417063±6123 | **8.87** |
| LS++ | | | **2.2779E+07** | 2.2781E+07±824 | 13.93 |
| FLS | | | **2.2779E+07** | 2.2780E+07±769 | 142.15 |
| LLOYD | SPNET_3D(434,874*3) | OOM | 2.2781E+07 | 2.2782E+07±609 | **0.49** |
| MLSP | | | **2.2779E+07** | **2.2779E+07±0** | 121.03 |
| MLS | | | **2.2779E+07** | 2.2781E+07±1022 | **10.33** |
| LS++ | | | **3.8720E+06** | 3.8722E+06±108.8 | 37.1 |
| FLS | | | **3.8720E+06** | 3.8721E+06±44.1 | 338.19 |
| LLOYD | syn(1,000,000*2) | OOM | **3.8720E+06** | 3.8724E+06±236.4 | **0.69** |
| MLSP | | | **3.8720E+06** | **3.8721E+06±32.9** | 223.97 |
| MLS | | | **3.8720E+06** | 3.8722E+06±119.2 | **22.93** |
| LS++ | | | **6.9094E+08** | **6.9094E+08±0** | 286.41 |
| FLS | | | **6.9094E+08** | **6.9094E+08±0** | 3244.76 |
| LLOYD | USC_1990(2,458,685*68) | OOM | **6.9094E+08** | **6.9094E+08±0** | **18.57** |
| MLSP | | | **6.9094E+08** | **6.9094E+08±0** | 2443.28 |
| MLS | | | **6.9094E+08** | **6.9094E+08±0** | **227.59** |
| LS++ | | | **4.6299E+07** | 4.7891E+07±1.6E+06 | 305.27 |
| FLS | | | **4.6299E+07** | 4.8013E+07±1.5E+06 | 3894.45 |
| LLOYD | SUSY(5,000,000*17) | OOM | **4.6299E+07** | 4.8261E+07±1.3E+06 | **29.2** |
| MLSP | | | **4.6299E+07** | **4.7349E+07±7.7E+04** | 2610.89 |
| MLS | | | **4.6299E+07** | 4.7612E+07±1.6E+06 | **262.61** |
| LS++ | | | **2.2614E+08** | 2.2726E+08±1.1E+06 | 614.07 |
| FLS | | | **2.2614E+08** | 2.2720E+08±9.2E+05 | 8728.13 |
| LLOYD | HIGGS(11,000,000*27) | OOM | **2.2614E+08** | 2.2858E+08±2.6E+06 | **109.06** |
| MLSP | | | **2.2614E+08** | **2.2718E+08±8.8E+05** | 6492.59 |
| MLS | | | **2.2614E+08** | 2.2728E+08±1.2E+06 | **522.15** |
| LS++ | | | 1.5774E+13 | 1.6098E+13±1.6E+11 | 31994 |
| FLS | | | **1.5649E+13** | 1.5856E+13±1.1E+11 | 18306 |
| LLOYD | SIFT(100,000,000*128) | OOM | 1.5718E+13 | **1.5793E+13±1.5E+11** | **401** |
| MLSP | | | - | - | >48h |
| MLS | | | 1.5718E+13 | 1.5848E+13±1.1E+11 | **12738** |

Table 18: Comparison results on clustering costs and running time with $k = 3$ on datasets with sizes larger than $50,000$, where OOM is short for out of memory

| Method | Size | BB(Cost) | Best | Mean | Time(s) |
|---|---|---|---|---|---|
| LS++ | | | 284.27 | 287.39±2.78 | 3.34 |
| FLS | | | 290.15 | 290.15±0 | 18.86 |
| LLOYD | rds(50,000*3) | 282.65(4h) | 284.22 | 287.50±8.67 | **0.86** |
| MLSP | | | **283.13** | **283.27±0.21** | 58.61 |
| MLS | | | 284.21 | 287.23±2.72 | **2.97** |
| LS++ | | | **1.9201E+08** | 1.9205E+08±2.1E+04 | 2.17 |
| FLS | | | **1.9201E+08** | 1.9205E+08±1.6E+04 | 10.92 |
| LLOYD | KEGG(53,413*23) | 1.92E+08(3901s) | **1.9201E+08** | 1.9754E+08±1.6E+07 | **0.26** |
| MLSP | | | **1.9201E+08** | **1.9201E+08±0** | 24.83 |
| MLS | | | **1.9201E+08** | 1.9204E+08±2.3E+04 | **1.88** |
| LS++ | | | **5.62E+04** | 56281±76 | 5.33 |
| FLS | | | **5.62E+04** | 56248±49 | 26.22 |
| LLOYD | Urban_10(100,000*2) | 56231(4h) | **5.62E+04** | 57768±2762 | **0.14** |
| MLSP | | | **5.62E+04** | **56246±36** | 40.97 |
| MLS | | | **5.62E+04** | 56315±83 | **4.64** |
| LS++ | | | **5.0721E+14** | 5.0721E+14±3.3E+11 | 15.65 |
| FLS | | | **5.0721E+14** | 5.0739E+14±3.0E+11 | 122.09 |
| LLOYD | RNG_AGR(199,843*7) | 5.0721E+14(4h) | **5.0721E+14** | 5.0781E+14±3.2E+11 | **0.35** |
| MLSP | | | **5.0721E+14** | **5.0736E+14±3.1E+11** | 73.2 |
| MLS | | | **5.0721E+14** | 5.0759E+14±3.0E+11 | **14.24** |
| LS++ | | | **201878** | 202554±954 | 16.48 |
| FLS | | | **201878** | 201879±2.5 | 97.19 |
| LLOYD | Urban_GB(360,177*2) | OOM | **201878** | 204943±7816 | **0.53** |
| MLSP | | | **201873** | **201874±2.03** | 103.94 |
| MLS | | | **201878** | 202166±629 | **14.89** |
| LS++ | | | 8.8280E+06 | 8.8309E+06±1.4E+03 | 26.85 |
| FLS | | | 8.8280E+06 | 8.8289E+06±1.1E+03 | 331.24 |
| LLOYD | SPNET_3D(434,874*3) | OOM | 8.8280E+06 | 8.8315E+06±1.6E+03 | **0.83** |
| MLSP | | | **8.8273E+06** | **8.8273E+06±0.17** | 180.12 |
| MLS | | | 8.8277E+07 | 8.8313E+06±1.0E+03 | **22.94** |
| LS++ | | | 1.8662E+06 | 1.8696E+06±3.2E+03 | 66.78 |
| FLS | | | 1.8765E+06 | 1.8674E+06±1.8E+03 | 527.22 |
| LLOYD | syn(1,000,000*2) | OOM | 1.8661E+06 | 1.9579E+06±1.1E+03 | **0.85** |
| MLSP | | | **1.8652E+06** | **1.8659E+06±1.1E+03** | 391.86 |
| MLS | | | 1.8658E+06 | 1.8694E+06±3.1E+03 | **31.62** |
| LS++ | | | **3.9017E+08** | 3.9375E+08±1.E+07 | 439.74 |
| FLS | | | **3.9017E+08** | **3.9017E+08±0** | 4802.6 |
| LLOYD | USC_1990(2,458,685*68) | OOM | **3.9017E+08** | 3.9861E+08±3.6E+07 | **20.11** |
| MLSP | | | **3.9017E+08** | **3.9017E+08±0** | 3612.99 |
| MLS | | | **3.9017E+08** | 3.9732E+08±1.4E+07 | **324.97** |
| LS++ | | | 4.0623E+07 | 4.1035E+07±1.5E+06 | 451.28 |
| FLS | | | 3.9880E+07 | 4.0997E+07±4.5E+04 | 5329.33 |
| LLOYD | SUSY(5,000,000*17) | OOM | 3.9848E+07 | 4.0648E+07±6.4E+05 | **31.67** |
| MLSP | | | **3.8720E+07** | **3.9316E+07±1.4E+05** | 4137.19 |
| MLS | | | 3.9904E+07 | 4.0882E+07±3.6E+06 | **297.43** |
| LS++ | | | **2.0829E+08** | 2.1016E+08±1.3E+06 | 1316.59 |
| FLS | | | **2.0829E+08** | 2.1012E+08±1.2E+06 | 18429.3 |
| LLOYD | HIGGS(11,000,000*27) | OOM | **2.0829E+08** | 2.1046E+08±5.7E+06 | **114.18** |
| MLSP | | | **2.0829E+08** | **2.1010E+08±1.1E+06** | 12713.2 |
| MLS | | | **2.0829E+08** | 2.1018E+08±1.4E+06 | **1019.6** |
| LS++ | | | 1.5035E+13 | 1.5194E+13±8.8E+10 | 41376 |
| FLS | | | **1.4778E+13** | 1.4994E+13±9.1E+10 | 27908 |
| LLOYD | SIFT(100,000,000*128) | OOM | 1.4805E+13 | 1.5021E+13±8.0E+10 | **531** |
| MLSP | | | - | - | >48h |
| MLS | | | **1.4778E+13** | **1.4992E+13±8.9E+10** | **12931** |

Table 19: Comparison results on clustering costs and running time with $k = 5$ on datasets with sizes larger than $50,000$, where OOM is short for out of memory

## B.4 Experiments on Different Local Search Algorithms with Fixed Time Limits

In this section, we conduct some additional experiments on different datasets with fixed time limits. The experimental results in Table 20 demonstrate that our proposed MLSP algorithm achieves the best clustering quality with fixed time limits.

| Dataset | Method | Time Point | Cost | Time Point | Cost | Time Point | Cost | Time Point | Cost |
|---|---|---|---|---|---|---|---|---|---|
| rds | LS++ | | 137.96 | | 137.96 | | 137.93 | | 137.93 |
| | FLS | | 138.98 | | 137.26 | | 134.68 | | 134.62 |
| | MLSP | 20s | **133.62** | 40s | **132.29** | 60s | **132.28** | 80s | **132.14** |
| | MLS | | 140.19 | | 138.23 | | 137.63 | | 137.63 |
| KEGG | LS++ | | 62855194 | | 62855194 | | 62793065 | | 62793065 |
| | FLS | | 61633499 | | 61633499 | | 61633499 | | 61608744 |
| | MLSP | 20s | **61534479** | 40s | **61534479** | 60s | **61534479** | 80s | **61534479** |
| | MLS | | 63419827 | | 63419827 | | 63419827 | | 63419827 |
| Urban_10 | LS++ | | 25011.62 | | 25011.62 | | 25011.62 | | 25011.62 |
| | FLS | | 25103.83 | | 25066.53 | | 25058.21 | | 24973.42 |
| | MLSP | 20s | **24767.85** | 40s | **24767.66** | 60s | **24751.62** | 80s | **24748.37** |
| | MLS | | 24920.38 | | 24885.80 | | 24885.80 | | 24885.80 |
| RNG_AGR | LS++ | | 1.3833E+14 | | 1.3833E+14 | | 1.3833E+14 | | 1.3833E+14 |
| | FLS | | 1.3822E+14 | | 1.3813E+14 | | 1.3800E+14 | | 1.3797E+14 |
| | MLSP | 60s | **1.3743E+14** | 120s | **1.3703E+14** | 180 | **1.3690E+14** | 240s | **1.3689E+14** |
| | MLS | | 1.3826E+14 | | 1.3826E+14 | | 1.3826E+14 | | 1.3826E+14 |
| Urban_GB | LS++ | | 89234.23 | | 89234.23 | | 89234.23 | | 89234.23 |
| | FLS | | 89691.46 | | 89691.46 | | 89688.77 | | 89406.39 |
| | MLSP | 60s | 89314.46 | 120s | 89311.78 | 180 | **88946.64** | 240s | **88407.99** |
| | MLS | | **88955.80** | | **88955.80** | | 88955.80 | | 88955.80 |
| SPNET_3D | LS++ | | 2576396 | | 2576396 | | 2575784 | | 2575784 |
| | FLS | | 2573606 | | 2572637 | | 2571369 | | 2570776 |
| | MLSP | 200s | **2570513** | 400s | **2570292** | 600s | **2570160** | 800s | **2569484** |
| | MLS | | 2574266 | | 2574266 | | 2574266 | | 2574266 |
| syn | LS++ | | 561444.2 | | 561444.2 | | 561444.2 | | 561444.2 |
| | FLS | | 561914.9 | | 561914.9 | | 561914.9 | | 561914.9 |
| | MLSP | 300s | **560502.5** | 600s | **560502.4** | 900s | **560502.4** | 1200s | **560502.4** |
| | MLS | | 562385.8 | | 562385.8 | | 562385.8 | | 562385.8 |
| USC_1990 | LS++ | | 270910366 | | 270910366 | | 270910366 | | 270910366 |
| | FLS | | 270910366 | | 270910366 | | 270910366 | | 270825616 |
| | MLSP | 2000s | **270735773** | 4000s | **270735773** | 6000s | **270735773** | 8000s | **270735773** |
| | MLS | | 270825616 | | 270825616 | | 270825616 | | 270825616 |
| SUSY | LS++ | | 32437956 | | 32437956 | | 32437956 | | 32437956 |
| | FLS | | **31693128** | | 31693128 | | 31693128 | | 31693128 |
| | MLSP | 2500s | 31756413 | 5000s | **31650548** | 7500s | **31650548** | 10000s | **31629757** |
| | MLS | | 32548680 | | 32548680 | | 32548680 | | 32548680 |
| HIGGS | LS++ | | 189640888.6 | | 188373729.5 | | 188373729.5 | | 187098096.4 |
| | FLS | | 189186932.5 | | 189186932.5 | | 189186932.5 | | 189186932.5 |
| | MLSP | 12000s | **184084129.5** | 24000s | **184084129.5** | 36000s | **184084129.5** | 48000s | **184084129.5** |
| | MLS | | 186689935.5 | | 186689935.5 | | 186689935.5 | | 186689935.5 |
| SIFT | LS++ | | 1.82E+13 | | 1.77E+13 | | 1.67E+13 | | 1.58E+13 |
| | FLS | | 1.79E+13 | | 1.59E+13 | | 1.55E+13 | | 1.52E+13 |
| | MLSP | 20000s | 1.85E+13 | 40000s | 1.85E+13 | 60000s | 1.85E+13 | 80000s | 1.85E+13 |
| | MLS | | **1.38E+13** | | **1.38E+13** | | **1.38E+13** | | **1.38E+13** |

Table 20: Comparison results on clustering costs with $k = 10$ and fixed time limits

