# OpenReview forum: "Linear Time Algorithms for k-means with Multi-Swap Local Search"
_NeurIPS.cc/2023/Conference — NeurIPS 2023 poster_

### Official Review · Reviewer_bF7j · 2023-06-25

**Soundness:** 3 good
**Presentation:** 4 excellent
**Contribution:** 3 good
**Rating:** 7
**Confidence:** 3

**Summary:**

This paper studies local-search algorithms for k-means clustering. The goal here is to obtain a local-search algorithm which (1) give a constant factor approximation and (2) run in time linear in the dataset. In the past literature, one can distinguish essentially 2 types of local-search algorithms for this problem. Single-swap local search which have a simple swap procedure but can only guarantee approximation ratios around 500 at least. Multi-swap local search which use a richer swap structure to give better guarantees (the current best is 9+\epsilon – approx) but are often impractical since finding a good swap often implies to enumerate over all subsets of some candidate set, which makes the algorithm slower than linear (in the dataset size) time.

This paper tries to reconcile these two directions by giving a new Multi-swap local search algorithm which gives and approximation factor of 50+\epsilon and runs in essentially linear time. One of the main ideas in this work is to use sampling techniques in the spirit of the famous k-means++ algorithm and only after this try to swap current centers with the sampled candidates. Because the set of sampled candidates is much smaller than the actual dataset, the running time is significantly improved. They also give experiments that show that their algorithm performs well in practice.


**Strengths:**

In my opinion, the strengths of the paper are as follows.
1)	Interesting result and techniques. Not being an expert, it seems that it is the first time that the sampling technique is used in the context of multi-swap local search.
2)	The algorithm seems to perform well in practice. Experimental section is quite thorough.


**Weaknesses:**

Some minor weakness in my opinion is that there is still a significant gap between the lower bound of 9 for local search and the proven upper bounds.

Typos :
line 88: benefit
line 105: much *more*
line 145: I guess the OPT value is missing in the inequality


**Questions:**

The authors actually propose 2 algorithms, first MLS for which they prove theoretical guarantees. And a second MLPS which they say is “more practical” but I do not see anywhere in the paper that the same approximation guarantee of 50 holds for MLPS as well. In the experiment sections, MLS seems to perform well, but it is unclear to me if this is the same MLS as the one that was analyzed in section 3 (line 336 the authors say they use the sampling method of MLPS for the MLS algorithm). Does this change the theoretical guarantee of MLS? It would be great of the authors could clarify these points.


**Limitations:**

yes

---

> ### Author Rebuttal · Authors · 2023-08-09
>
> We thank the reviewer for the positive rating and the thoughtful comments. In the following we address the concerns.
>
> **Question 1: The authors actually propose 2 algorithms, first MLS for which they prove theoretical guarantees. And a second MLSP which they say is "more practical" but I do not see anywhere in the paper that the same approximation guarantee of 50 holds for MLSP as well. In the experiment sections, MLS seems to perform well, but it is unclear to me if this is the same MLS as the one that was analyzed in section 3 (line 336 the authors say they use the sampling method of MLSP for the MLS algorithm). Does this change the theoretical guarantee of MLS? It would be great of the authors could clarify these points**
>
> Response: We thank the reviewer for raising this important question. The proposed MLSP algorithm is actually a heuristic algorithm for achieving better practical performance in experiments. The main heuristic strategies for MLSP are: 1) random sampling for accelerating the clustering cost updating process; 2) recombination for finding potentially better solutions. As a result, the theoretical bound $50(1+\frac{1}{t}) + \epsilon$ might not be applicable to MLSP.
>
> The proposed theoretical guarantee for MLS algorithm is based on the standard worst-case analysis. However, in practical scenarios, heuristic strategies can help to improve both the clustering quality and runtime of the proposed algorithm. For the MLS algorithm used in experiments, as stated in line 336-337, the sampling strategy is also used to accelerate the clustering cost updating process. Thus, the bound of $50(1+\frac{1}{t}) + \epsilon$ might not hold in this case.
>
> In experiments, it can be seen from table 2 that the running time of MLS and MLSP algorithm can be accelerated by using heurstic strategies such that they can achieve better performance compared with other local search methods, and MLSP performs much better than MLS.
>
> How to give a theoretical bound on the approximation guarantee for the MLSP algorithm is an interesting problem that deserves further study.

---

> > ### Comment · Reviewer_bF7j · 2023-08-14
> >
> > I would like to thank the authors for their response and clarification. After looking at other reviews and corresponding rebuttals, my initial assessment remains. I think this is an interesting paper.

---

### Official Review · Reviewer_hjYC · 2023-07-04

**Soundness:** 3 good
**Presentation:** 2 fair
**Contribution:** 3 good
**Rating:** 6
**Confidence:** 5

**Summary:**

The authors study the well-known k-means problem: Given a set of points in the Euclidean space, compute k centers such that the sum of squared distances between points and their closest center is minimized. A constant-factor approximation to this problem is achieved by local search: Start with arbitrary k centers and improve the solution by choosing a point which is currently not a center and replace a center in our current solution by this point. If we exchange at most t centers instead of one in the local search method Kanungo et al. showed that the algorithm computes a (3+2/t)^2-approximation.

Lattanzi and Sohler proposed a combination of k-means++ sampling and local search (LS++): Start with the k centers computed by the k-means++ solution and improve it via local search. In every step of local search the new center is sampled proportionally to its current cost in the solution. Lattanzi and Sohler showed that their algorithm computes a 509-approximation after O(k*log(log(k))) local search steps.

The authors extend the algorithm of Lattanzi and Sohler (MLS): Instead of sampling only one point, sample t points S simultaneously and search for a subset T of S such that replacing |T| points in the current solution by T reduces the cost significantly. The main result is the following: After O(k^O(t)*log(epsilon*log(k))) local search steps the algorithm returns a (50(1+1/t)+epsilon)-approximation in expectation. Each local search step takes time O(n*d*k^O(t)), where n is the number of points and d is the dimension of the euclidean space.

In the experimental part of the paper the authors implemented the MLS-algorithm for t=2 and proposed a second algorithm (MLSP): 1) Use k-means++ for an initial solution 2) For R rounds do the following in each round: Apply the MLS-algorithm, check whether the current solution C is the best observed so far. If yes, again apply MLS to improve it, if not exchange some (or all) centers from C according to a procedure developed by the authors and apply MLS with new center set.
So MLSP restarts MLS with a new set of centers if the local optimum which MLS computes is not good. This restart is performed R times. The authors compare the MLS and MLSP algorithm to three algorithms LS++, Lloyd and FLS on several data-sets (50 000- 11 000 000 points) for k=10. In most cases the MLSP algorithm computes the best solution but has a significantly higher run-time than MLS, Lloyd and LS++.

**Strengths:**

The extension of LS++ to an algorithm which samples more than one center is natural and the resulting improvement in the approximation factor is significant. Even though some parts of the proof are similar to the approach of Lattanzi, Sohler and Kanungo et al. the proof involves some non-trivial steps and is original in my opinion.

The experiments show that the algorithm MLSP outperforms other algorithms with respect to the quality of the solution even in the setting where all algorithms run for a fixed amount of time. This may be due to the restart of the local search procedure when it runs into a bad local minimum, so it seems to be a good heuristic to implement in any local search procedure.

**Weaknesses:**

Especially the technical part of the paper is very hard to read, which could have been prevented since it is mostly not due to the difficulty of the proofs but because of the complex notation and sometimes imprecise statements. While the result in the main theorem seems to be plausible I am not completely sure about the correctness of the proof.

It would be nice to have experiments for k>10, since the choice of k<=10 seems not to fit the considered data sizes of 50 000 to 10 000 000 points.  The run-time of MLS and MLSP rises faster with respect to k than the run-time of LS++ so it would be nice to see how their algorithms perform for larger values of k.

Furthermore the authors outline that the MLS algorithm is relatively fast, especially when compared to its easier version LS++ where only one center is exchanged in every step, which was confusing at first. However the speed-up in the run-time is achieved by an approximate computation of the clustering cost of a solution which could have also been implemented for LS++ for a fair comparison to MLS in my opinion.

**Questions:**

L 69: Should this be n^t instead of n*k^t? If not, please explain. (appears also in L 76, Table 1)
L 88: Missing O(...) in the run-time
L 123: I feel that you should state Lemma 2 as a theorem
L 136-138: You should mention that this only holds when the cost of the current solution is larger than ((50+1/t)+epsilon)*OPT.
L 138: we can get an approximation -> we get an approximation
L 145: In the lower bound on the cost of C there is missing the cost of an optimal solution.
L 167: Maybe just introduce a map \phi which assigns every center in C* the nearest center in C? Then you do not need the notation s_c and you can denote the set of optimal centers captured by a center c in C by \phi^{-1}(c). I feel that the notation should be simplified at a lot of places throughout the paper.
L  172: find a set \sigma_h of unused lonely centers -> let \sigma_h be an arbitrary set of unused lonely centers
L 173: in the definition of A: don't we have c_h^* in \phi(s_c_h^*) already by definition as c_h^* is captured by s_c_h^*?
L 176-177: Please specify how you construct type 2 matched swap pairs. At first it seems that every unused lonely center from C should be in at most one type 2 matched swap pair, but this is not possible. I think what you later need is that every unused lonely center and every c_h^* with |\phi(s_c_h^*)|>t should form a type 2 matched swap pair?
L 179-182: For the union of clusters with centers in V you use X(V) while for the union of clusters with centers in Q you use Z(J(Q)), this is very confusing. I suggest to simplify the notation (maybe just directly write the union of clusters instead of introducing a new notation) and also don't change between center sets, clusters sets and union of cluster sets that much in the whole paper when it is avoidable.
L 212-222: maybe it is better to state this as a lemma
L 219-220: missing two " ) " in this chain of inequalities, also please put this into an align environment as it is hard to read if you include it directly in the text
L 227:  cost of those optimal clusters -> cost of optimal clusters
L 230: |Q|>1 : Isn't this automatically the case for sets in H_2?
L 235: c_h^* can find clustering centers close to it -> c_h^* is close to a center from C
L242: you use H_1,H_2  etc. are sets of centers, now H_2^* is some set of clusters, this is a little bit confusing
L267: Shouldn‘t the probability Omega(1/k) also depend on t because of the definition in Line 231?
L268-269: I belive that this should be independency of events, not the union bound
L281: comma at beginning of line
L281: Could you explain why we add Q_S‘‘ and not Q_T to H_G?
Algorithm 3 Line 11: Could you explain how this swapping exactly works? Also why don‘t use simply Lloyd here?

Appendix:
L428: In the following sequence of equations: first inequality remember the reader why s_{o_p} is not in V
L446: rename one of the (c_h^*,c_h^m) in the argmin equation
L448: \mu is previously used as centroid of a set, maybe find an other letter
L449: In the following sequence of equations, first inequality: should \mu(L) be \mu(M_3)? If not please define \mu(L)
L452: Please explain this upper bound
L460: please state the bound for the relaxed triangle inequality as lemma somewhere
L463: In the following sequence of equations: Z(Q_T) → Z(J(Q_T)), Z(Q‘_S)-Z(J(Q‘_S)), Z(Q)->Z(J(Q))
L474-476: It is not clear if you mean the failure probability of not sampling at least one point correctly or all points correctly
L484: In the following inequality: shouldn‘t „t“ be „|v(W)|“ in the exponent?
L488 In the following inequalities, last inequality: \Delta(P,C)→ 300\Delta(P,C)
L496: „unique lonely center“ misleading as one could think that l(c_h^*) is specific to P_h^*
L515: \lambda t^{-k} → \lambda k^{-t}
L520-528: this paragraph equals L512-520 → remove

**Limitations:**

n.a.

---

> ### Author Rebuttal · Authors · 2023-08-09
>
> We thank the reviewer for the positive rating and the thoughtful comments. In the following we address the concerns.
>
> **Q1: Regarding $n^t$ instead of $nk^t$ in L69**
>
> Response: The swap size $O(nk^t)$ used in L69 is actually $O((nk)^t)$, which is a typo. Sorry for the confusion.
>
> **Q2: Typos and misleading statements**
>
> Response: We thank the reviewer for pointing these out and we apologize for being unable to respond to each typo error and misleading statement individually due to the character limitations in the rebuttal. In our revised version, we we will carefully correct all the typos and misleading statements in the paper.
>
>
> **Q3: Maybe just introduce a map $\phi$ which assigns every center in $C^{*}$ the nearest center in $C$?**
>
> Response: Thanks for the great  suggestion. To make the notations simpler and easier to follow, in our revised version, we will introduce a mapping function $\phi$ to map every center in $C^*$ to the nearest center in $C$, and use $\phi^{-1}(c)$ to denote the set of centers captured by $c$.
>
> **Q4: Regarding the definition of $A$ in L173**
>
> Response: Thanks for pointing this out. In the construction of $A$, $c_h^*$ already belongs to $\Psi(s_{c_h^*})$ and $\Psi(s_{c_h^*}) \cup \{c_h^*\}$ is indeed $\Psi(s_{c_h^*})$.
>
> **Q5: L176-177, please specify how you construct type 2 matched swap pairs**
>
> Response: We are sorry for the confusion caused by the statements on construction of type 2 matched swap pairs. In L176, every unused lonely center should participate in the construction of type 2 swap pairs with every $s_{c_h^*}$ satisfying the condition of $|\Psi(s_{c_h^*})|>t$. In our revised version, we will make the statement clearer
>
> **Q6: Regarding the notations in L179-182**
>
> Response: We thank the reviewer for the kind suggestions. In the revised version, we will make the notation for the union of clusters with centers in $V$ and $Q$ consistent.
>
> **Q7: L230, $|Q|>1$. Isn't this automatically the case for sets in $H_2$**
>
> Response: $|Q|>1$ is implicit in the definition of $H_2$. We will remove it in the revised version.
>
> **Q8: Regarding the notations in L242**
>
> Response: We are sorry for the confusion caused by the notations. In our revised version, we will make the notations consistent.
>
> **Q9: Regarding the probability bound in L267**
>
> Response: We thank the reviewer for pointing this out. Since the swap size $t$ is usually a constant and could be much smaller than $k$, we omit the dependence on $t$ in the probability lower bound. In the revised version, we will add $t$ to the lower bound.
>
> **Q10: L281: Could you explain why we add $Q_S''$ and not $Q_T$ to $H_G$**
>
> Response: Thanks for raising this question. $H_G$ is denoted as the collection of optimal clusters whose clustering cost is relatively small with respect to $C$. Recall that $Q_T = Q_S'' \cup Q_L$. For each $P_h^* \in Q_L$, with good probability, data points close to $c_h^*$ can be sampled in the $m(P_h^*)$-iteration of the independent $t$ sampling iterations. Thus, $Q_S''$ is added to $H_G$ while $Q_L$ is not added to $H_G$.
>
> **Q11: Regarding L11 in Algorithm3**
>
> Response: We thank the reviewer for raising this important question. In L11 of Algorithm3, the swapping works as follows. For each center $c_h \in C$, the algorithm finds the nearest 50 data points in $P$ to it (denoted as the set $N(c_h)$). For each point $x \in N(c_h)$, a swap pair $(c_h, x)$ is constructed. If one of the swap pairs constructed can induce a clustering cost reduction, the algorithm will execute the swap. In L11 of Algorithm3, the nearest neighbor searching and the swap pairs construction processes will be repeated until the current clustering reaches a convergence.
>
> The reason that Lloyd's method is not used in L11 is to prevent the algorithm from falling into a poor local optimum too early. Once Lloyd's method is used to adjust the centers, a local optimum is obtained, which reduces the possibility to find potentially better solutions.
>
> **Q12: Regarding the notations in L449**
>
> Response: The $\mu(L)$ term appears in the first inequality actually refers to $L$, which is a typo. Sorry for the confusion.
>
> **Q13: L452: Please explain this upper bound**
>
> Response: We thank the reviewer for raising this question. In the following, we give a brief explanation on how to obtain the upper bound in L452, which will appear in the revised version. Let $L1 = \cup_{c_h^* \in L}s_{c_h^*}$. By the definition of $\mu(M_3)$ and $L$, for each $s_{c_h^*} \in L1$, we can find a set $z(s_{c_h^*}) \subseteq \mu(M_3)$ with size $|\Psi(s_{c_h^*})| - 1$ such that $z(a) \cap z(b) = \emptyset$ for any $a, b \in L1$. For each $s_{c_h^*} \in L1$, since $|\Psi(s_{c_h^*})| > t$, $|\Psi(s_{c_h^*})|/|z(s_{c_h^*})| \le 1 + 1/t$. Taking a summation for centers in $L$, we have $|L| = \sum_{c_h^* \in L}1 = \sum_{s_{c_h^*} \in L1} |\Psi(s_{c_h^*})| \le \sum_{s_{c_h^*} \in L1} |z(s_{c_h^*})|(1+1/t) \le |\mu(M_3)|(1+1/t)$.
>
> **Q14: Regarding the relaxed triangle inequality in L460**
>
> Response: We thank the reviewer for the nice suggestion. In our revised version, we will include a lemma to state the relaxed triangle inequality in the preliminary section.
>
> **Q15: Regarding the failure probability in L474-476**
>
> Response: We thank the reviewer for pointing this out. To make the statement clearer, we will rewrite it as "The probability of failing to sample a point close to each of the optimal clustering centers".
>
> **Q16: L484 about $|\nu(W)|$ in the exponent**
>
> Response: We thank the reviewer for pointing this out. The reason why $t$ is in the exponent instead of $|\nu(W)|$ is based on the following relation between $t$ and $|\nu(W)|$.
>
> By the definition of $\nu(W)$, we have $\nu(W) = \lbrace{P_h^*:P_h^* \in Q''_S\rbrace}$ and $Q''_S$ is a subset of $Q$. Note that $Q \in H_2$. By the definition of $H_2$, we have $|Q| \le t$. Thus, $|\nu(W)| \le t$ and $t$ can be used to replace $|\nu(W)|$ in the exponent of the probability bound.

---

> > ### Comment · Reviewer_hjYC · 2023-08-16
> >
> > Thank you for the detailed response.

---

### Official Review · Reviewer_dZuQ · 2023-07-08

**Soundness:** 3 good
**Presentation:** 3 good
**Contribution:** 3 good
**Rating:** 6
**Confidence:** 3

**Summary:**

Local search algorithm is a well studied technique for clustering problems. In the k-means problem, the simple swap heuristic states that start with an intiial chosen set of k centers  and then at each local search step, check if swapping an existing center with a new one leads to decrease in cost. It is known that this gives a constant factor approximation algorithm (with the constant being 25). The $t$-swap heuristic generalises this local search algorithm by swapping a subset of $t$ centers at each step. It is clear that the running time of such an algorithm will be exponential in $t$, though the approximation factor converges to 9 as $t$ increases.

A related line of work on the k-means problem deals with the Lloyd's algorithm. In this algorithm, we choose an initial set of k seed centers and then run a different local search algorithm on it. A lot of work has happened on how to choose these initial centers. The k-means++ heuristic says that we choose the initial centers as follows: the first center is picked uniformly at random. After having picked $i$ initial centers, a point is chosen as the $(i+1)^{th}$ one with probability proportional to square of the distance from the $i$ centers chosen already. It is known that if we choose a set of $k$ centers in this manner, then it can lead to $O(\log k)$-approximation algorithm for the k-means objective (and this bound is tight). Subsequent works have focussed on whether we can improve this initial set in a small amount of time. For example, reference [10] showed that if we run the single swap heuristic on the $k$ centers chosen by this random sampling procedure for about $O(k)$ steps, then we get a 509-approximation algorithm.

This paper is an extension of this last result by giving a fast implementation of the $t$-swap heuristic on an initial set of centers chosen using the random sampling procure described above. Each local search step chooses a set of $t$ new centers using a random sampling procedure and checks if it can replace an existing set of $t$ centers in the current solution. The paper shows that the running time now becomes $O(k^{O(t)} nd$, and hence is linear in $n$ for fixed $k,t$. They also show that the resulting set of centers has approximation ratio of about 50. The analysis has the same structure as that of [10], though analyzing $t$-swap is much more tricky, and requires new insights.

**Strengths:**

1. Usually $t$-swap heuristcs are expensive to implement, this paper gives a non-trivial implementation which is efficient in practice.
2. The analysis of $t$-swap requires new ideas.

**Weaknesses:**

1. The improvement in experiments is only marginal (in most cases less than 5%).

2. The usual local search heurstic (with single swap heuristic), ie.., reference [9] in table 1, has running time $O(nkd \lof \Delta)$ and has approximation ratio of 25. This running time almost matches the running time of the result in this paper (upto log \Delta factor), and has better approximation ratio. So it is not clear if the theoretical contribution is significant. I also don't see a comparison with [9] (with single swap heuristic) in the implementation.

**Questions:**

1. Please compare with [9] in terms of theoretical and experimental results.
2. It may be worth explaining what new ideas are needed in the analysis as compared to [10].

**Limitations:**

No negative impact.

---

> ### Author Rebuttal · Authors · 2023-08-09
>
> We thank the reviewer for the positive rating and the thoughtful comments. In the following we address the concerns.
>
> **Question 1: Please compare with [9] in terms of theoretical and experimental results**
>
> Response: We thank the reviewer for raising this question. We are sorry for the confusion caused by the typo on the time complexity of the algorithm in [9]. The term of $O(n^{t}k^{t}d\log\Delta)$ in Table 1 should be $O(n^{t+1}k^{t}d\log\Delta)$.
>
> Our proposed MLS algorithm can achieve a $(50(1+\frac{1}{t})+\epsilon)$-approximation in time $(ndk^{2t+1}\log(\epsilon^{-1}\log k))$. It can be seen that even with a single swap strategy ($t=1$), the algorithm in [9] (denoted as LS algorithm for short) has at least quadratic running time in the data size while the running time of our proposed MLS algorithm has a linear dependence on the data size. For the LS algorithm, it is not surprising to get a better ratio with quadratic running time.
>
> For experimental performance, as shown in [8] (the FLS algorithm), even with single-swap strategy, the LS algorithm can hardly handle datasets with size over 10,000 due to its quadratic time complexity. Thus, we did not include the results of LS algorithm for comparison in our experiments. In the following table, we give additional results (see table 1) to compare our MLS and MLSP algorithms with the LS algorithm on different datasets used in our experiments with size smaller than 10,000. To ensure a fair comparison, we maintain a consistent number of 400 local search iterations, as employed in our previous experiments. It can be seen that our proposed MLSP algorithms outperform LS algorithm in terms of both clustering cost and running time. On average, the average clustering cost is reduced by 1.15\% compared with LS using our MLSP algorithm while the running time of our MLSP algorithm is more than 1000 times faster than LS algorithm.
>
> **Table 1: Comparison results of LS, MLS and MLSG on datasets with size smaller than 10,000**
> |Iris(150*4)|Best Cost|Average with Std|Time(s)|Abs_FL(720*21)|Best Cost|Average with Std|Time(s)|
> |:---:|:---:|:---:|:---:|:---:|:---:|:---:|:---:|
> |LS|29.79|30.001$\pm$0.23|300.61|LS|**1.0786E+06**|1.0873E+06$\pm$7.8E+03|1513.83|
> |MLSP|**29.74**|**29.761$\pm$0.03**|0.43|MLSP|**1.0786E+06**|**1.0786E+06$\pm$0**|0.91|
> |MLS|29.93|30.214$\pm$0.19|**0.17**|MLS|**1.0789E+06**|1.0970E+06$\pm$1.1E+04|**0.47**|
> |**SEEDS(210*7)**||||**TR(980*10)**||||
> |LS|214.951|218.9401$\pm$5.79|436.54|LS|**762.16**|767.88$\pm$4.1|2049|
> |MLSP|**214.523**|**215.079$\pm$0.49**|0.63|MLSP|**762.16**|**764.71$\pm$1.9**|0.94|
> |MLS|214.954|219.294$\pm$2.82|**0.46**|MLS|785.01|805.89$\pm$11.5|0.56|
> |**GLASS(214*9)**||||**SGC(1000*21)**||||
> |LS|**251.859**|252.9835$\pm$0.91|424.28|LS|1.1741E+08|1.2026E+08$\pm$3.5E+06|2146.91|
> |MLSP|**251.859**|**251.994$\pm$0.54**|0.52|MLSP|**1.1734E+08**|**1.1752E+08$\pm$5.1E+04**|9.39|
> |MLS|253.294|254.001$\pm$0.55|**0.46**|MLS|1.1735E+08|1.1846E+08$\pm$9.6E+04|**0.62**|
> |**BM(249*6)**||||**HEMI(1955*7)**||||
> |LS|**375974**|378169$\pm$2739.33|528.74|LS|2.7073E+06|2.7346E+06$\pm$5.7E+05|3713.27|
> |MLSP|**375974**|**376276$\pm$265.69**|0.52|MLSP|**2.7070E+06**|**2.7123E+06$\pm$6.7E+03**|1.08|
> |MLS|378649|384982$\pm$4131.91|**0.37**|MLS|2.7292E+06|2.7886E+06$\pm$6.9E+04|**0.59**|
> |**UK(258*5)**||||**pr2392(2392*2)**||||
> |LS|**29.268**|29.602$\pm$0.34|538.55|LS|**5.3578E+09**|5.4745E+09$\pm$9.0E+08|4958.77|
> |MLSP|**29.268**|**29.286$\pm$0.01**|0.61|MLSP|**5.3578E+09**|**5.3668E+09$\pm$2.3E+07**|1.16|
> |MLS|29.411|30.004$\pm$0.27|**0.39**|MLS|5.3629E+09|5.4203E+09$\pm$6.2E+07|**0.58**|
> |**HF(299*12)**||||**TRR(5456*24)**||||
> |LS|**6.9604E+10**|7.2964E+10$\pm$4.1E+09|634.03|LS|**1.3796E+05**|1.3834E+05$\pm$566.8|12646|
> |MLSP|**6.9604E+10**|**6.9604E+10$\pm$0**|0.71|MLSP|**1.3796E+05**|**1.3829E+05$\pm$253.1**|2.99|
> |MLS|**6.9604E+10**|7.0148E+10$\pm$4.3E+08|**0.4**|MLS|1.4591E+05|1.4939E+05$\pm$2.0E+03|**0.64**|
> |**WHO(440*8)**||||**AC(7195*22)**||||
> |LS|**3.3631E+10**|3.3764E+10$\pm$1.7E+08|913.23|LS|**1163.7**|1169.19$\pm$8.42|18036.1|
> |MLSP|**3.3631E+10**|**3.3676E+10$\pm$4.6E+07**|0.89|MLSP|**1163.7**|**1168.2$\pm$9.01**|1.51|
> |MLS|3.4324E+10|3.5212E+10$\pm$5.6E+08|**0.4**|MLS|1165.5|1181.7$\pm$12.8|**0.64**|
> |**HCV(572*12)**||||**rds_cnt(10000*4)**||||
> |LS|1.1312E+06|1.1458E+06$\pm$4.4E+03|1311.96|LS|1.6104E+06|1.6364E+06$\pm$2.3E+04|20544.7|
> |MLSP|**1.1311E+06**|**1.1410E+06$\pm$2.2E+04**|0.9|MLSP|**1.6099E+06**|1.6105E+06$\pm$7.2E+02|0.88|
> |MLS|1.1505E+06|1.2135E+06$\pm$4.2E+04|**0.49**|MLS|1.6146E+06|1.6520E+06$\pm$2.5E+05|**0.42**|
>
> **Question 2: It may be worth explaining what new ideas are needed in the analysis as compared to [10]**
>
> Response: Thanks for the great suggestion. For the LS++ method with single swap strategy in [10], its theoretical guarantee relies heavily on the one-to-one matched swap pairs constructed. When directly applying the ideas in [10] to multi-swap analysis, it has the following challenges: (1) the theoretical bounds on the clustering cost after multi-swaps can not be established; (2) for multi-swap local search, a successful swap involves that data points close to each center of a subset of optimal clustering centers should be simultaneously sampled.
>
> However, the LS++ method can only guarantee that a data point close to a single optimal clustering center can be sampled with high probability. To establish the bounds of clustering cost after multi-swaps, we extend the notions of swap pairs to swap sets and propose a new consecutive sampling method to construct candidate centers for swap. To analyze the success probability in each local search step, we propose new structures that divide optimal clusters into different groups for establishing a lower bound of sampling success probability. In summary, new definitions for swap pairs and new structures for analyzing the success probability lower bounds are needed in the analysis as compared to [10].

---

### Official Review · Reviewer_BGQc · 2023-07-24

**Soundness:** 3 good
**Presentation:** 3 good
**Contribution:** 2 fair
**Rating:** 6
**Confidence:** 4

**Summary:**

This paper proposes a multi-swap local search algorithm for the k-means problem with linear running time in the data size, while also achieves a better approximation ratio when compared with other local search algorithms that adopt single-swap strategy. To benefit more from such algorithm when handling large-scale datasets, the authors further propose a sampling-based method to accelerate the updating of clustering cost during the swaps. The extensive empirical experiments validate the good numerical performance of the proposed algorithms.

**Strengths:**

This paper studies the classic k-means clustering problem and gives a first linear time local search algorithm in literature that adopts the multi-swap strategy. Such algorithm enjoys the two advantage from both worlds: smaller approximation ratio and linear run-time in data size. The organization of this paper is clear and well-structured, the mathematical proof is very detailed and easy to follow, and the numerical experiments are also quite extensive.

**Weaknesses:**

For Lloyd algorithm, according to your experiments it is able to save up to 99% of the run time (0.07 vs 5.12), while only having around 1% more average cost. Therefore, the merit carried by the work presented in this paper does not seem to be too impressive from the practical point of view. One related question is, considering the huge amount of time being saved from Lloyd algorithm, have the authors tried running Lloyd algorithm for multiple times (if Lloyd algorithm takes 0.07s for each initialization, then you can run 5.12/0.07 times of Lloyd algorithm) with different initialization and then simply take the best solution? How does that compare with MLS when they are set to have the same run time?

**Questions:**

Line 11-12, "which improves the current best result 509": It might be true for local search methods that take single-swap strategy, but It is unfair to call this the best result, since the current best approximation ratio is 6.357.

Line 90, "MLS algorithm": Please refer it and write out the full name when mentioning this abbreviation for the first time. Please consider move the "multi-swap local search algorithm" at line 100 to here.

Algorithm 1: "Output: ... of at most k centers". I do not understand why you say "at most". Since in each iteration point p can be selected if and only if point p is not in C, isn't it guaranteed that C has size exactly k? Similar question to Algorithm 2.

Line 3, Algorithm 1: $\Delta$ is defined over two sets, here p is a point it should be {p}, and {C} should be C. This appears in other places as well, for instance Algorithm 2.

In the numerical experiment section, in Table 2 why is MLS able to outperform the single-swap local search methods LS++ and FLS by a big margin? It does not make sense to me since MLS algorithm is clearly more complex than LS++ in [10] and it has bigger runtime complexity.

Line 345-346, "the average clustering ... nearly match the results of the BB method": As far as I can see, except on one dataset, isn't MLSP algorithm always outperform BB method in terms of average clustering costs?

**Limitations:**

The authors did not address the limitation of their work.

---

> ### Author Rebuttal · Authors · 2023-08-09
>
> We thank the reviewer for the positive rating and the thoughtful comments. In the following we address the concerns.
>
> **Question 1: Line 11-12, "which improves the current best result 509": It might be true for local search methods that take single-swap strategy, but It is unfair to call this the best result, since the current best approximation ratio is 6.357**
>
> Response: We thank the reviewer for pointing out this  issue. In the revised version of the paper, we will rewrite the statement as "which improves the current best result 509 with a linear running time in the data size."
>
> **Question 2: Line 90, "MLS algorithm": Please refer it and write out the full name when mentioning this abbreviation for the first time. Please consider move the "multi-swap local search algorithm" at line 100 to here.**
>
> Response: We thank the reviewer for the kind suggestion. In the revised version, we will give the full name of "MLS algorithm" when mentioning this abbreviation for the first time.
>
> **Question 3: Algorithm 1: "Output: ... of at most k centers". I do not understand why you say "at most". Since in each iteration point p can be selected if and only if point p is not in C, isn't it guaranteed that C has size exactly k? Similar question to Algorithm 2**
>
> Response: We thank the reviewer for raising this subtle issue.
>
> In the standard definition of the $k$-means problem, the goal is to output a set $C \subseteq \mathbb{R}^d$ with size at most $k$ such that the clustering cost is minimized. To be consistent with the literature, we phrase our output requirement as "a set $C \subseteq \mathbb{R}^d$ with size at most $k$". Since local search methods for the $k$-means problem in literature all return a solution of size exact $k$, we also follow the tradition and require our algorithms to return a solution of size exact $k$.
>
> **Question 4: Line 3, Algorithm 1: $\Delta$ is defined over two sets, here $p$ is a point it should be $\lbrace p \rbrace$, and $\lbrace C \rbrace$ should be $C$. This appears in other places as well, for instance Algorithm 2**
>
> Response:  Thanks for pointing this out. The $\Delta(p,\lbrace C \rbrace)$ used in line 3 of Algorithm 1 is actually $\Delta(\lbrace{p\rbrace},C)$, which is a typo. Sorry for the confusion.
>
> **Question 5: In the numerical experiment section, in Table 2 why is MLS able to outperform the single-swap local search methods LS++ and FLS by a big margin? It does not make sense to me since MLS algorithm is clearly more complex than LS++ in [10] and it has bigger runtime complexity**
>
> Response: We thank the reviewer for raising this important question. Theoretically, the running time of the MLS algorithm is larger than that of LS++ and FLS. The key point for MLS to outperform  LS++ and FLS in experiments is that the sampling-based method is applied to accelerate the swapping and clustering cost updating processes (as mentioned in line 337).
>
> **Question 6: Line 345-346, "the average clustering ... nearly match the results of the BB method": As far as I can see, except on one dataset, isn't MLSP algorithm always outperform BB method in terms of average clustering costs?**
>
> Response: Thanks for raising this question. Since the average clustering cost of our MLSP algorithm does not outperform that of the BB method on dataset RNG\_AGR, for the sake of rigor, we did not state that our proposed MLSP algorithm outperform the results of the BB method. In the revised version, we will make a clearer statement as "the average clustering costs outperform the results of the BB method except on one dataset RNG\_AGR".

---

### Author Rebuttal · Authors · 2023-08-10

We thank the reviewers for their positive feedback and valuable suggestions and we highly appreciate the effort paid by the reviewers to provide in-depth reviews that helped us to improve our work. In the following, we will address the reviewers' comments in detail with separate responses.

---

> ### Comment · Area_Chair_3mu4 · 2023-08-18
>
> Dear authors,
>
> Thank you for your rebuttal, and your messages.
> The reviewers and myself will take all of your points into consideration when making our decision.
>
> Best regards.

---

### Decision · Program_Chairs · 2023-09-21

**Decision:**

Accept (poster)

**Comment:**

This paper proposes a linear-time, local-search based, algorithm for k-means. All reviewers have a positive view of the paper.
I thus recommend it for acceptance.